# Metabotropic signaling within somatostatin interneurons controls transient thalamocortical inputs during development

Deepanjali Dwivedi [1,2,5], Dimitri Dumontier [3,5], Mia Sherer [1,2], Sherry Lin [1], Andrea M. C. Mirow[1,2,3], Yanjie Qiu [1,2], Qing Xu[1,2,4], Samuel A. Liebman[3], Djeckby Joseph[3], Sandeep R. Datta[1], Gord Fishell [1,2] ✉ & Gabrielle Pouchelon [1,2,3] ✉

During brain development, neural circuits undergo major activity-dependent restructuring. Circuit wiring mainly occurs through synaptic strengthening following the Hebbian "fire together, wire together" precept. However, select connections, essential for circuit development, are transient. They are effectively connected early in development, but strongly diminish during maturation. The mechanisms by which transient connectivity recedes are unknown. To investigate this process, we characterize transient thalamocortical inputs, which depress onto somatostatin inhibitory interneurons during development, by employing optogenetics, chemogenetics, transcriptomics and CRISPR-based strategies in mice. We demonstrate that in contrast to typical activity-dependent mechanisms, transient thalamocortical connectivity onto somatostatin interneurons is non-canonical and involves metabotropic signaling. Specifically, metabotropic-mediated transcription, of guidance molecules in particular, supports the elimination of this connectivity. Remarkably, we found that this process impacts the development of normal exploratory behaviors of adult mice.

Bottom-up afferents from the thalamus provide topographical precise sensory inputs to the neocortex. These sensory inputs target both excitatory and inhibitory cortical neuron populations and undergo dramatic refinement during development. It is well accepted that both genetic and activity-dependent factors are required for appropriate circuit development, but at present it is unclear how these factors interact to recalibrate thalamocortical (TC) circuits for adult function[1–3].

In the adult somatosensory cortex, feedforward inhibition (FFI) from TC inputs onto parvalbumin (PV) inhibitory cortical interneurons (cINs), which inhibit excitatory neurons, and feedback inhibition from excitatory neurons to somatostatin (SST) cINs play a crucial role in regulating adult circuit function[4–6]. The developmental pattern of connectivity expressed by TC afferents with their neuron targets, the

FFI, supports the broad principle that weak connections are strengthened over time during circuit maturation[7,8]. Specifically, TC afferents only weakly contact excitatory pyramidal cells and PV cINs during the first postnatal week of development and are later potentiated[7,9]. This developmental plasticity is thought to follow a Hebbian mechanism, in which "cells that fire together, wire together"[10–12]. The extent to which these are universal mechanisms or specific to these strengthening processes remains unclear. Remarkably, during the early postnatal week of development, SST cINs transiently receive TC inputs. While the mechanisms underlying transient connectivity are unknown, this early connection is involved in the TC potentiation onto the other cortical neurons[13,14]. As such, SST cINs, which are the earliest-born cINs populations, are thought to

[1]Harvard Medical School, Department of Neurobiology, Boston, MA, USA. [2]Broad Institute, Stanley Center for Psychiatric Research, Cambridge, MA, USA. [3]Cold Spring Harbor Laboratory, Cold Spring Harbor, Harbor, NY, USA. [4]Center for Genomics & Systems Biology, New York University Abu Dhabi, Abu Dhabi, UAE. [5]These authors contributed equally: Deepanjali Dwivedi, Dimitri Dumontier. ✉e-mail: gordon_fishell@hms.harvard.edu; pouchel@cshl.edu

orchestrate cortical network synchrony during development[15–17]. Here we explore the mechanisms by which TC inputs weaken onto SST cINs and the molecular players that mediate this process.

In contrast to the means by which TC inputs mature onto PV and excitatory neurons in the somatosensory cortex[12,18], we discovered that transient connectivity to SST cINs involves non-canonical activity-dependent mechanisms and that this process is modulated by metabotropic signaling. We identified the developmental postsynaptic expression of the metabotropic glutamatergic receptor 1 (mGluR1) in SST cINs specifically, as an important player for the TC input transient refinement through downstream gene transcriptional regulation, such as semaphorin 3A (Sema3A). These results suggest that postsynaptic SST cINs actively regulate circuit development by providing feedback signals, which ultimately underlie exploratory behaviors in adult mice.

## Results

### Input from VB neurons in the thalamus connect to SST cINs during early postnatal development

TC afferents to SST cINs in layer 5 (L5) of the primary somatosensory cortex (S1) are strong during the first postnatal week of development but substantially weakened as maturation proceeds[13,14]. Active sensory exploration in adulthood occurs through distinct thalamocortical neurons. During the first postnatal week, SST cINs primarily receive TC inputs from the VB neuron population in the thalamus, which conveys the primary sensory information[19] and recent work shows that SST cINs contribute to early, fast sensory-driven information transfer prior to active sensory exploration[16]. These results suggest that TC inputs from VB neurons ($TC_{VB}$) transiently contact SST cINs. To explore this hypothesis, we examined $TC_{VB}$ input selectivity onto SST cINs during postnatal development using optogenetic stimulation of $TC_{VB}$ terminals in L5 of S1 and measured their strength onto SST cINs (Fig. 1a). This was accomplished using a mouse driver line specific for VB thalamic neurons (Vipr2-Cre[20,21]) to activate Cre-mediated AAV-driven opsins (Fig. 1a-b). The monosynaptic excitatory postsynaptic currents (EPSCs) in SST cINs, optogenetically evoked by $TC_{VB}$, were observed to decrease over development. During early postnatal stages P6–7, which we define as the immature stage, $TC_{VB}$ inputs evoked larger EPSCs onto SST cINs than those observed onto this same population during more mature windows, P9–11 and P28-30 (Fig. 1c and Supplementary Fig. 1a), while their strength increases over time onto excitatory neurons as expected (Supplementary Fig. 1b). As strengthening during development is commonly associated with pruning and synapse elimination, we next examined the synaptic organization underlying transient dynamics. Using pan-TC pre- and postsynaptic markers (VGluT2[22] and Homer1, respectively), we observed that the density of TC synaptic contact follows the respective physiological strength of TC inputs onto SST and PV neurons in adulthood (Supplementary Fig. 1c, d). We examined TC synaptic contact density during development and found that the overall density of TC synaptic contacts onto SST cINs were consistently decreased in L5 (P5, P7, P10, and P30; Fig. 1d and Supplementary Fig. 1e, f). Together, both anatomical and functional data showed that the maturation of transient connectivity is associated with the weakening and synaptic refinement of $TC_{VB}$ inputs onto SST cINs.

### Postsynaptic activity of SST cINs regulates the maturation of transient TC inputs

The development of TC projections and cortical topographic maps have been shown to be activity-dependent[2,23,24]. More specifically, Hebbian mechanisms, in which "cells that fire together, wire together", are thought to underlie the developmental strengthening of TC inputs onto excitatory neurons[10–12]. In addition, presynaptic activity from $TC_{VB}$ inputs specifically instructs the development of their postsynaptic excitatory neuron target[25]. However, the above experiments indicate that $TC_{VB}$ synapses are distinct on SST cINs from the other cell types, suggesting a postsynaptic-specific mechanism for transient

connectivity. Therefore, we next investigated whether the postsynaptic activity of SST cINs controls TC input refinement. We hypothesized that, similar to Hebbian-based developmental potentiation[9], increasing postsynaptic activity would maintain strong TC inputs onto SST neurons, while decreasing it would promote the normal TC input weakening. Accordingly, we used designer receptors exclusively activated by designer drugs (DREADD)[26,27] and the Clozapine-N-oxide (CNO), to either activate (Gq) or inhibit (Gi) SST cINs during the first postnatal week (P0–P8; Fig. 2a and Supplementary Fig. 2a)[28,29] and determine how they affected the strength of $TC_{VB}$ inputs onto SST cINs (Fig. 2b). Contrary to our expectations, at the early mature stage, when the strength of TC inputs onto SST cINs is normally diminished, DREADD-Gi inhibition within SST cINs failed to decrease the strength of $TC_{VB}$ inputs. Instead, DREADD-Gq activation promoted the normal weakening. (Fig. 2c, d and Supplementary Fig. 2b). In addition, Gq(+) SST cINs exhibit a decrease in the density of TC synaptic contacts (VGluT2+/Homer1+), compared to Gi(+) and control SST cINs (Fig. 2e and Supplementary Fig. 2c, d). Notably, neither DREADD-manipulations significantly affected the developmental levels of apoptosis of SST cINs (Supplementary Fig. 2e)[28,30] and their intrinsic properties only revealed limited excitability modifications that did not affect their typical electrophysiological identity (Table S1)[31]. After CNO discontinuation (P8), these effects proved temporary, as $TC_{VB}$ inputs were no more reduced onto Gq(+) SST cINs at the later mature stage and instead displayed a mild rebound effect (Supplementary Fig. 2f). We further examined activity-dependent $TC_{VB}$ input maturation onto SST cINs using a well-established genetic tool, the Kir2.1 hyperpolarizing potassium channel. While Kir2.1 inhibition was previously shown to disrupt TC input refinement and cortical map formation[24,32–35], the chronic hyperpolarization of SST cINs during the first postnatal weeks did not significantly affect $TC_{VB}$ transient inputs, similar to the chemogenetic Gi-manipulation (Supplementary Fig. 2g–i). Interestingly, the absence of effect on the excitability of SST cINs upon DREADD-Gq manipulation, in contrast to DREADD-Gi and the hyperpolarizing Kir2.1, suggests that the mechanism involved in the regulation of transient connectivity is non-canonical, through metabotropic signaling rather than postsynaptic neuron excitability.

Altogether, these results suggest that TC transient connectivity onto SST cINs do not involve standard activity-dependent mechanisms during postnatal development, wherein Gq signaling promotes the normal depression of TC inputs.

### mGluR1 is highly expressed in SST cINs during development

We hypothesize that a metabotropic Gq signaling, endogenous to SST cINs, comparable to Gq-DREADD, might account for the normal weakening in TC afferents to SST cINs during development. We examined SST cIN-specific gene expression during the formation and maturation of TC transient connectivity. Specifically, we analyzed public single-cell RNA sequencing (scRNAseq) data from developing cINs collected during immature (at P2)[36] and early mature stages (at P10)[37]. We aligned the two datasets using canonical correlation analysis (CCA) followed by clustering using the Seurat pipeline[38,39]. We identified cIN types by curating the known inhibitory cIN markers in these cluster populations[37,40] and compared gene expression specific to SST cIN subtype clusters (Fig. 3a and Supplementary Fig. 3b) compared to PV or all other cIN types (Fig. 3b; Supplementary Fig. 3a; and Table S2). Among the three best markers of immature SST cINs was Grm1, a gene coding for the mGluR1, a Gq-coupled postsynaptic metabotropic glutamatergic receptor. mGluR1 expression peaks during the immature time window and persists in this population in mature stages (Fig. 3c–e and Table S3)[41–44]. Single-molecule fluorescent in situ hybridization (smFISH) confirmed the colocalization of Grm1 in SST cINs in L5 of S1 (Fig. 3f). Notably, mGluR1 expression was found to vary within discrete SST cIN subtypes, by P10, the early mature time point, in particular (Supplementary Fig. 3c, d). At both stages, the Myh8+ SST cIN

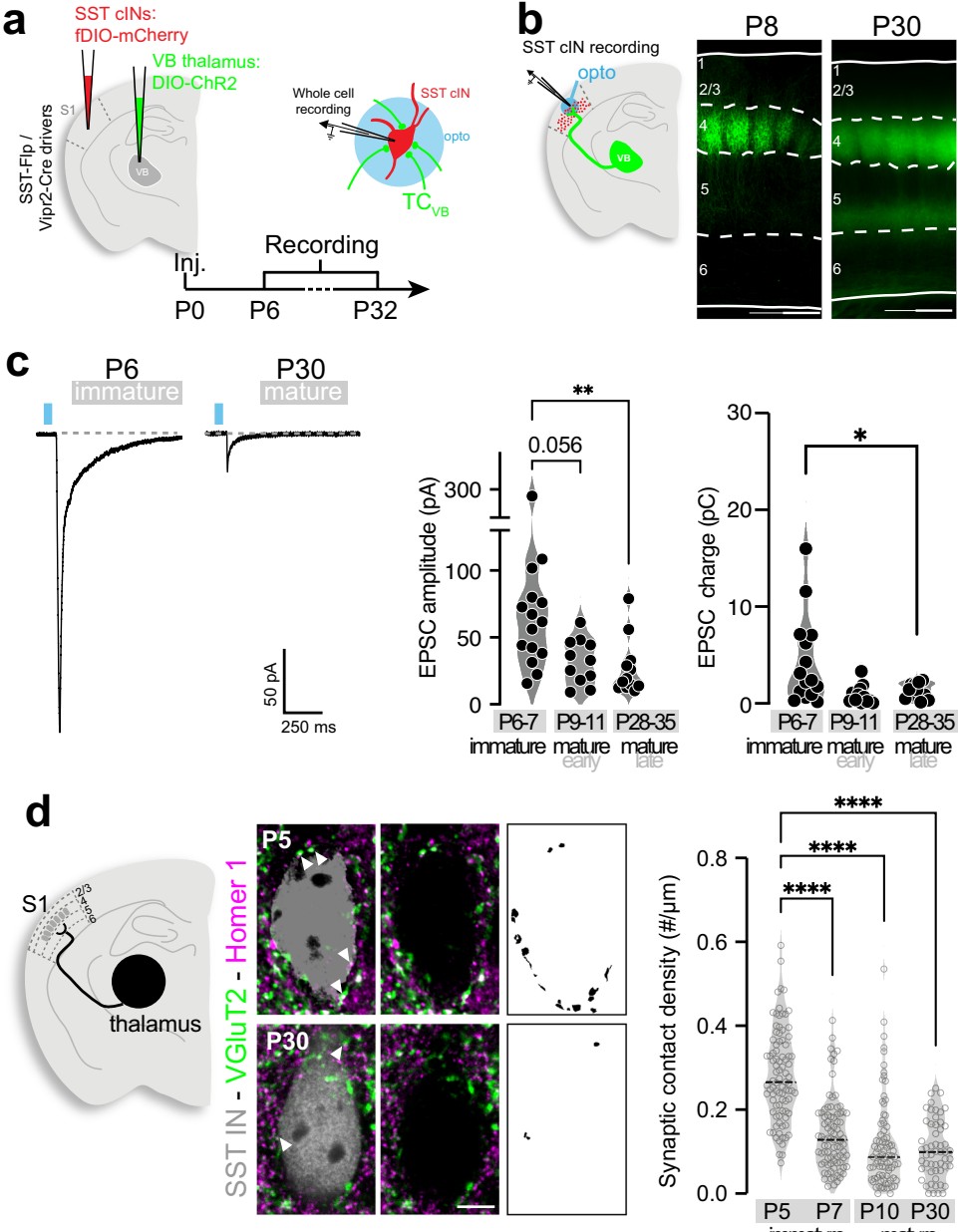

**Fig. 1 | TC inputs from VB transiently project onto SST cINs during postnatal development. a** AAV Cre-dependent DIO-ChR2(or -ChRmine) under the control of Vipr2-Cre, target VB thalamic neurons. Flp-dependent fluorescence (fDIO) reporters under the control of SST-FlpO target SST cINs. AAVs were injected at P0 and responses were recorded at multiple time points: immature (P6–P7), early mature (P9–11), and later mature stage (P28–35). **b** Example of $TC_{VB}$ projections pattern at P30 from ChR2 expression in VB. Scale bar 200 μm. **c** Average traces of unitary examples of SST cIN responses at P6 and P30 upon light stimulation of $TC_{VB}$ projections. The peak amplitude of EPSCs (excitatory postsynaptic currents) in L5 decreases over development (P6–7: 74 ± 12 pA, $n = 15$, $N = 4$; P9–10: 35.08 ± 4.67 pA, $n = 13$, $N = 3$; P28–35: 26.92 ± 6.01 pA, $n = 12$, $N = 3$; Kruskal–Wallis test $p = 0.0029$; Dunn's multiple comparison tests: P6 vs P10 $p = 0.056$; P6 vs P30 $p = 0.003$). EPSC charges in L5 also decrease over development (P6: 4.34 ± 1.18 pC, P10: 1.24 ± 0.21

pC, P30: 0.89 ± 0.28 pC; Kruskal–Wallis test $p = 0.0004$; Dunn's multiple comparison tests: P6 vs P30 $p = 0.0238$). **d** Staining of VGluT2 (presynaptic) and Homer1 (postsynaptic) TC synaptic contacts onto SST cINs, labeled with SST-cre mouse line and fluorescent reporters. Masks of puncta colocalizations on the soma at P5 and P30 in L5. Scale bar 5 μm. Quantification of TC synaptic contact density onto SST cINs in L5. During development, the number of contacts onto SST soma is decreased (P5: 0.282 ± 0.01, $n = 89$, $N = 3$; P7: 0.141 ± 0.009, $n = 87$, $N = 3$; P10: 0.115 ± 0.011, $n = 86$, $N = 2$; P30: 0.105 ± 0.009, $n = 54$, $N = 3$; Kruskal–Wallis test $p = 5.41E-25$; Dunn's multiple comparison tests: P5 vs P7 $p = 1.75E-12$, P5 vs P10 $p = 3.69E-16$, P5 vs P30 $p = 9.63E-21$, P7 vs P10 $p = 0.151$; P7 vs P30 $p = 0.290$; P10 vs P30 $p = 0.999$). Data were presented as mean ± SEM. The number of biologically independent animal replicates = $N$ and cell replicates = $n$. Source data are provided as a Source Data file.

population, which is found in L5 of the cortex, as well as the Chodl[+] population, exhibit the highest expression of Grm1. Remarkably, mGluR1 immunohistochemistry revealed significant dendritic distribution of the receptor, which persists throughout adulthood, while somatic expression is mainly apparent during the immature time window (P3-P5) (Fig. 3g and Supplementary Fig. 3e). The high mGluR1

level expression at P30 suggests that mGluR1 specificity in SST cINs decreases over time through the acquisition of mGluR1 expression in other cell types, as recently described at adulthood[44,45]. These data nominate mGluR1 as a candidate for triggering metabotropic postsynaptic signaling in SST cINs to control transient TC inputs during early postnatal development.

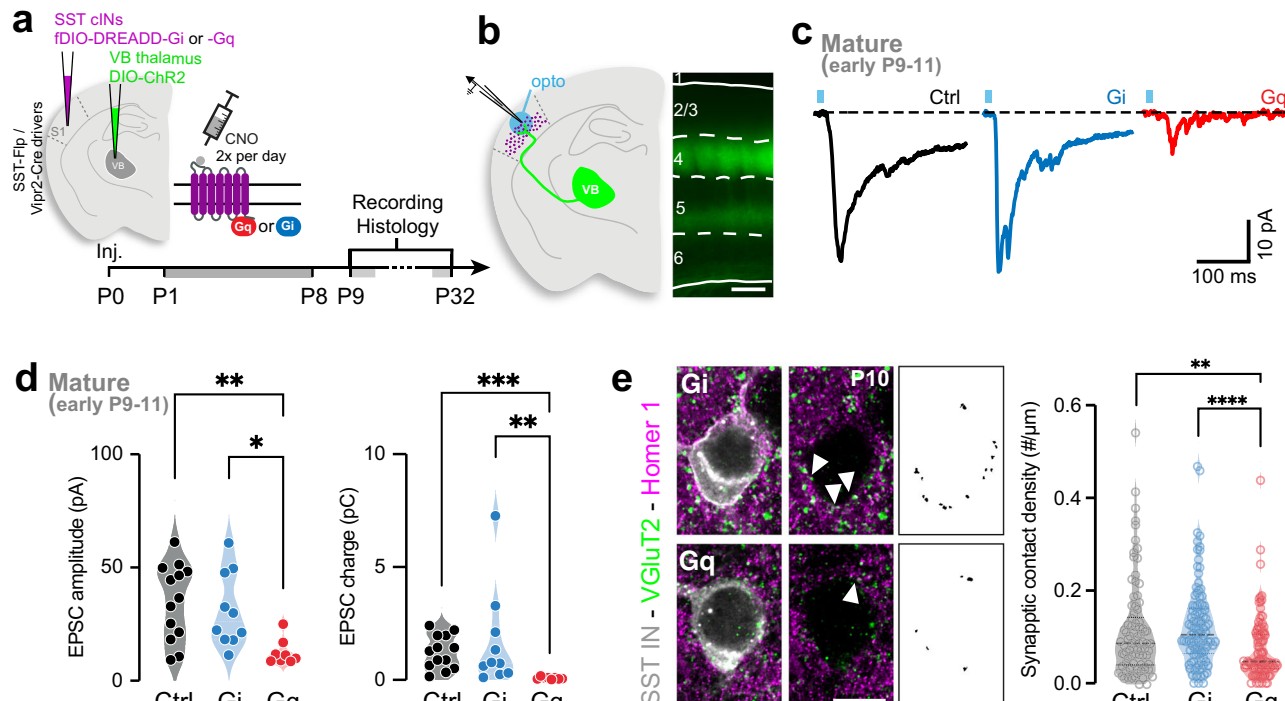

**Fig. 2 | Postsynaptic activity of SST cINs regulates the maturation of transient TC inputs. a** Designer receptors exclusively activated by designer drugs (DREADD)-based chemogenetic tools used to test activity-dependent maturation of TC inputs onto SST cINs. 12 to 24 h following AAV injections in the cortex and in the thalamus at P0, DREADD-Gi or -Gq are chronically activated with Clozapine-N-Oxide (CNO) injection twice a day until P8. Recording and synaptic analysis are performed after maturation during early and later mature windows (P9–11 and P28–32). **b** AAV-driven Cre-dependent ChR2 expression in TC$_{VB}$ inputs at P10. **c** Example average EPSCs from Ctrl, chronically inhibited with Gi or -activated with Gq SST cINs. **d** During the early mature window (P9–11), EPSC peak amplitudes and charges are significantly decreased after Gq expression compared to Control and Gi-expressing SST cINs (Amplitudes: Ctrl: 35.08 ± 4.67 pA, $n = 13$, $N = 3$ (includes Ctrl from Fig.1); Gi: 31.22 ± 5.17 pA, $n = 10$, $N = 3$; Gq: 12.68 ± 2.021 pA, $n = 8$, $N = 3$; Kruskal–Wallis test

$p = 0.0066$; Dunn's multiple comparison test: Ctrl vs Gq $p = 0.0038$, Gi vs Gq $p = 0.0222$; Charges, Ctrl: 1.24 ± 0.21 pC; Gi: 1.68 ± 0.69 pC; Gq: 0.08 ± 0.02; Kruskal–Wallis test $p = 0.0002$; Dunn's multiple comparison test: Ctrl vs Gq $p = 0.0003$; Ctrl vs Gi $p = 0.999$; Gi vs Gq $p = 0.0026$). **e** Staining of TC synaptic contacts (VgluT2) onto L5 SST cINs (Homer1). Scale bar: 10 μm. Quantification showing a decrease of the number of synaptic contacts onto SST cINs expressing Gq compared to SST cINs expressing Gi and to P10 Ctrl cells from Fig.1d (Ctrl: 0.116 ± 0.011, $n = 86$, $N = 2$; Gi: 0.123 ± 0.009, $n = 105$; $N = 3$; Gq 0.074 ± 0.007 $n = 94$ $N = 3$; Kruskal–Wallis test $p = 0.000016$; Dunn's multiple comparison tests: Ctrl vs Gq $p = 0.0078$, Ctrl vs Gi $p = 0.4542$, Gi vs Gq $p = 0.000011$). Data were presented as mean ± SEM. The number of biologically independent animal replicates = $N$ and cell replicates = $n$. Source data are provided as a Source Data file.

## mGluR1 CRISPR-deleted SST cINs retain their strong developmental TC inputs

mGluR1 is a postsynaptic metabotropic receptor primarily coupled with Gq[46]. Based on the high and specific postnatal expression of mGluR1 in SST cINs during postnatal development, we hypothesized that mGluR1-Gq signaling promotes the SST cIN-specific TC input refinement during development. To test this hypothesis, we investigated the function of mGluR1 in the weakening of TC transient connectivity using SST cIN-specific loss-of-function. To avoid non-cell autonomous effects of the mGluR1 deletion, from thalamic neurons in particular[47], we developed an AAV-based CRISPR strategy to knock down (KD) mGluR1 specifically from SST cINs in S1 (Fig. 4a). Grm1 smFISH coupled with SST immunostaining, confirmed the Grm1 transcript depletion from SST cINs at P5 and P10 using the CRISPR strategy (Fig. 4b and Supplementary Fig. 4a–c). mGluR1 KD did not significantly alter cellular density (Supplementary Fig. 4d) and the physiological properties of SST cINs at the early mature stage (Table S4). We next examined the maturation of TC$_{VB}$ inputs onto SST cINs in this model. We measured EPSCs from mGluR1 KD SST cINs in response to optogenetic stimulation of TC$_{VB}$ inputs (Fig. 4c). At this stage, EPSCs of KD SST cINs were larger than those recorded from control SST cINs (Fig. 4d and Supplementary Fig. 4e). The persistence of developmental TC inputs on SST cINs in absence of mGluR1 was corroborated by an increase of TC synaptic contact density, as labeled with VGluT2$^+$ and Homer1$^+$ onto KD SST cINs (Fig. 4e and

Supplementary Fig. 4f). The normal TC input weakening never occurred in absence of mGluR1, as EPSC evoked in SST cINs by TC$_{VB}$ remained immature through P30, a stage when S1 is mature (Supplementary Fig. 4g)[48]. These results reveal that TC synaptic maturation is regulated by mGluR1 in SST cIN during development.

## mGluR1-dependent transcriptional regulation in SST cINs regulates TC transient connectivity

mGluRs do not only modulate postsynaptic neuron activity, but have also been shown to regulate gene transcription[49] and translation[46,50]. We, therefore, hypothesized that activation of mGluR1 during the first postnatal week of development modulates transcriptional levels of genes mediating negative feedback signaling of TC inputs. We select four gene candidates, the function of which has been shown to be regulated by mGluR1 in the brain, and tested whether their transcriptional levels were regulated downstream of mGluR1 in postnatal SST cINs: two glutamate receptors, the subunit 3A of NMDA (Grin3a)[51–55] and GluD1 (Grid1)[56–58], and two guidance molecules, Semaphorin (Sema) 3A and 7A[59,60]. Using the integrated scRNAseq dataset, we observed that Grin3A, Grid1, and Sema3A are highly expressed in SST cINs at an immature stage, similar to the Grm1 expression pattern (Fig. 5a and Supplementary Fig. 5a–d). We examined the transcriptional levels of these genes in the SST cINs in the absence of mGluR1 using smFISH (Fig. 5b, c). In SST cIN deleted for mGluR1, using the same CRISPR strategy, we observed a downregulation in the RNA

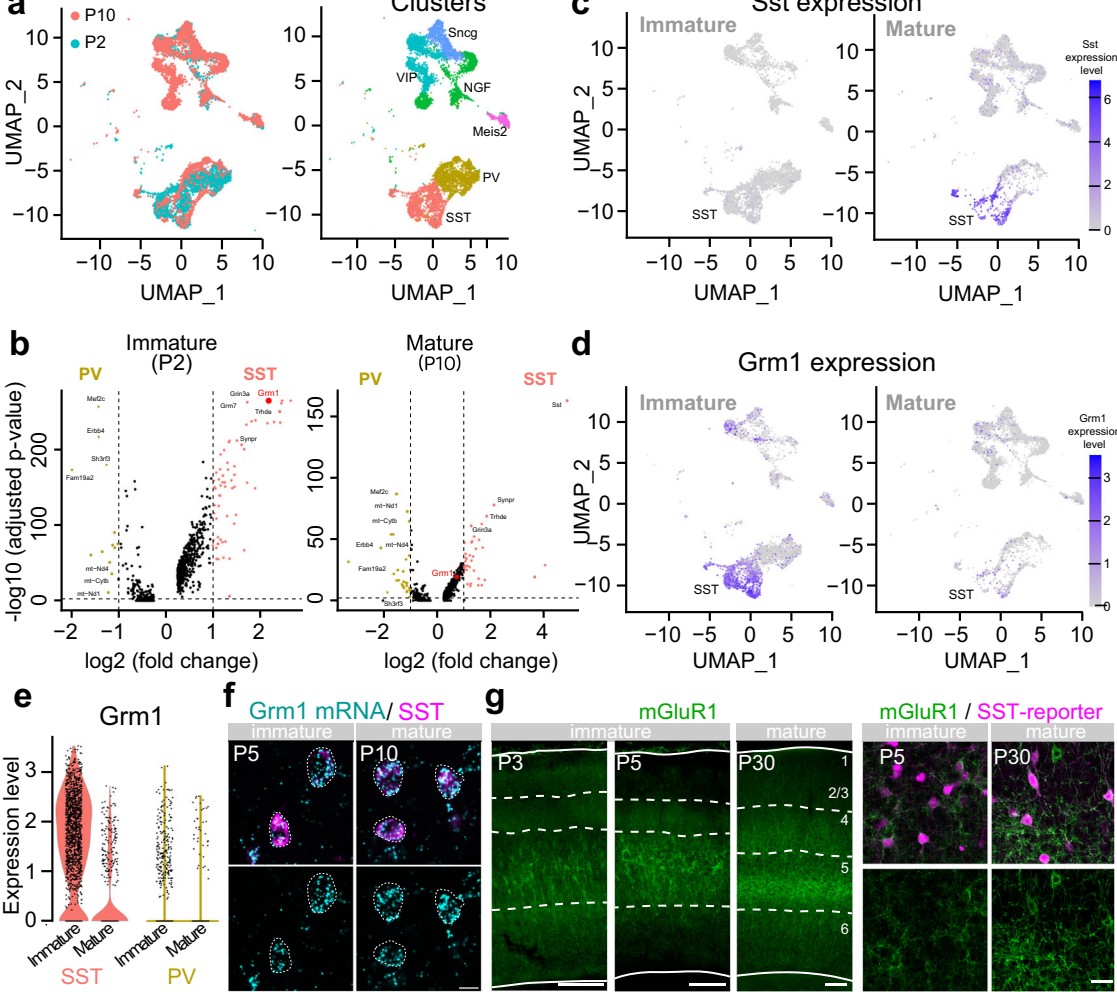

**Fig. 3 | mGluR1 is highly expressed in SST cINs during development. a** UMAP representation of the integration of P2 and P10 cortical GABAergic neurons scRNAseq public databases[36,37] (representing gene expression during the immature and early mature windows), allowing for the identification of SST cINs at both stages. The left UMAP is color-coded by time points. Right UMAP is color-coded by major cIN classes based on the expression markers as previously described[37,40]. **b** Differential gene expression between PV and SST cIN clusters using Seurat non-parametric two-sided Wilcoxon rank-sum test. Volcano plot representing the average of the Fold Change (FC) and adjusted *p* value of genes specifically expressed in PV and SST cIN clusters. During the immature stage (P2), Grm1 coding for mGluR1 is in one of the three highest significant genes in SST cINs and the top third differentially expressed gene (FC = 4.53, adjusted (adj.) *p* = 9e-264). While still

significant during the early mature stage (P10) (adj. *p* = 7.43e-20), the FC is only x1.64. **c**, **d** UMAP representation of Sst and Grm1 expression levels onto scRNAseq immature (P2) and mature (P10) datasets, showing the reverse temporal dynamics in expression levels. **e** Violin plots representing the level of Grm1 expression in cells from SST and PV cIN clusters at both immature and mature stages (P2/P10). **f** smFISH of Grm1 combined with SST immunostaining at P5 (*N* = 2) and P10 (*N* = 2) shows the high colocalization of Grm1 RNA molecules in SST cINs. Scale bar: 10 μm. **g** Immunofluorescence of mGluR1 in the S1 cortex (P5/P10 *N* = 1; P30 *N* = 5). Expression is high throughout development including both immature and mature time windows, and is associated with a soma-to-neurites redistribution of the receptor. Left scale bars: 100 μm; Right scale bar: 20 μm. Source data are provided as a Source Data file.

expression levels of Grin3A, Sema3A, and Sema7A, while Grid1 was upregulated (Fig. 5c–e and Supplementary Fig. 5e).

Sema3A is secreted and primarily acts as a repulsive guidance cue of axons, notably through Plexin A1 interactions[61–63]. Remarkably, the examination of the expression of Plexin A1 in the thalamus using the Allen in situ hybridization developing mouse brain atlas[64] revealed similar dynamics of Plexin A1 expression during postnatal development, as Sema3A (Supplementary Fig. 5f). Thus, in terms of both Sema3A in SST cINs and its cognate receptor in TC afferents, the timing of their expression is consistent with these functioning to mediate mGluR1-mediated downregulation of TC inputs to SST cINs. To further test this hypothesis, we examined whether Sema3A is necessary to control the maturation of TC transient connectivity onto SST cINs. To do so, we knocked down Sema3A from SST cINs using a similar CRISPR strategy as for targeting mGluR1 (Fig. 5f) and measured TC$_{VB}$ input strength onto Sema3A KD and Ctrl SST cINs (Fig. 5g). At the early

mature stage, both amplitude and charge distributions of Sema3A KD SST cINs responses were significantly enlarged compared to those from control neurons (Fig. 5h), suggesting a role for Sema3A in the maturation of transient connectivity. Altogether, these results show that mGluR1 regulates the expression of genes that function as feedback regulators of the transient connectivity between TC inputs and SST cINs.

## The persistence of developmental TC inputs alters exploratory behaviors at adulthood

Our TC$_{VB}$ data, as well as a recent study[16], suggest that TC synapses transiently provide inputs to SST cINs during a critical developmental window when sensory activity and SST cINs collaborate to regulate cortical circuit maturation and assembly. We, therefore, hypothesized that adult circuit functions as reflected by behavior, are altered in our model of early postnatal mGluR1-dependent SST cIN maturation

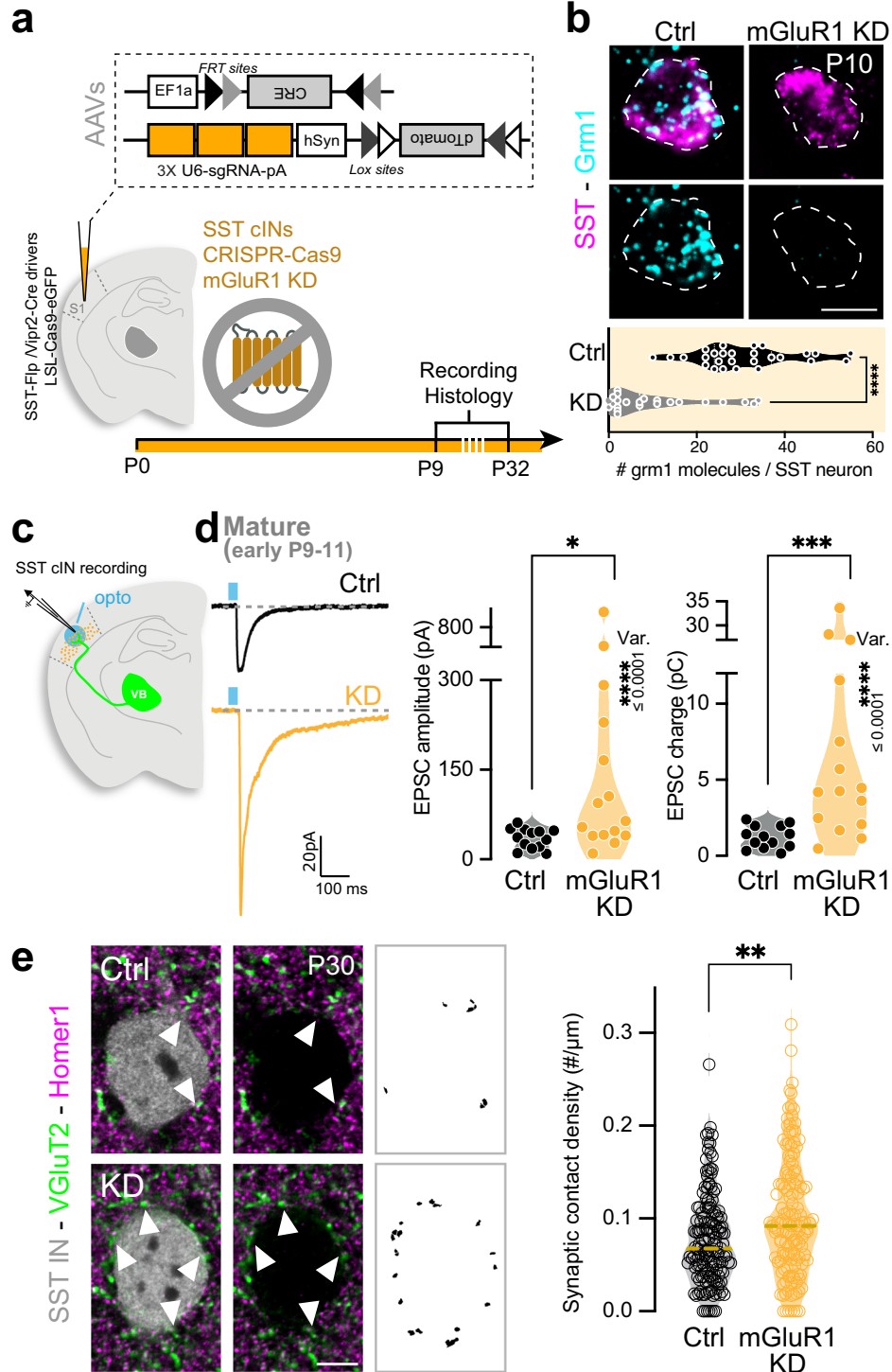

**Fig. 4 | mGluR1 CRISPR-deleted SST cINs retain their strong developmental TC inputs. a** Strategy for early deletion of mGluR1 (KD) in SST cINs specifically, using CRISPR/Cas9 system: AAV-driven Flp-dependent Cre expression is driven by SST-FlpO mouse line and controls LSL-Cas9eGFP mouse line. 3x CRISPR single guide RNAs targeting mGluR1 are driven by an AAV expressing a Cre-dependent reporter. AAVs were injected in S1 at P0. **b** Top: example of Grm1 expression in Ctrl and KD SST cINs using smFISH of Grm1 combined with SST immunostaining at P10. Scale bar: 10 μm. Bottom: Quantification of Grm1 RNA molecules per SST cINs using smFISH (Ctrl: 30.89 ± 2.17, $n = 27$, $N = 2$; KD: 10.35 ± 2.35, $n = 23$, $N = 2$; two-sided Student $t$-test $p = 5.89E\text{-}08$). **c** AAV-driven Cre-dependent ChR2 expression in VB is controlled by the Vipr2-Cre mouse line injected at P0. SST cINs were recorded in L5 during early and late mature windows (P9–11 and P29–32). **d** Averaged EPSCs from unitary examples of Ctrl and KD SST cINs at the early mature stage. EPSCs peak

amplitudes and charges are significantly increased in SST cINs deleted for mGluR1 compared to Ctrl at early mature stage (Amplitudes, Ctrl (from Fig. 1): 35.08 ± 4.67 pA, $n = 13$, $N = 3$; KD: 185.6 ± 66.4 pA, $n = 15$, $N = 4$; two-sided Mann–Whitney test $p = 0.0112$; Charges: Ctrl: 1.24 ± 0.21 pC, KD: 9.20 ± 2.84 pC; two-sided Mann–Whitney test $p = 0.0004$). The variance (Var.) of the responses is also significantly increased for KD SST cIN compared to Ctrl cells (Bartlett's test, Amplitudes: $p = 4.43E\text{-}12$; Charges: $p = 7.76E\text{-}12$). **e** Staining of TC synaptic contacts (VGluT2) onto L5 SST cINs (Homer1) at P30. Scale bar: 5 μm. Quantification showing that TC synapse density is increased onto mGluR1 KD SST cINs in L5 compared to Ctrl cells (Ctrl 0.077 ± 0.004, $n = 152$, $N = 4$; mGluR1 KD 0.101 ± 0.005, $n = 140$, $N = 4$; two-sided Mann–Whitney test $p = 0.0014$). Data were presented as mean ± SEM. The number of biologically independent animal replicates = $N$ and cell replicates = $n$. Source data are provided as a Source Data file.

within S1. The complex functions of adult SST cINs, normally controlling feedback inhibition[4–6], long-range motor and association input integration[65] prompted us to perform unsupervised behavioral analysis, focusing on pseudo-naturalistic open-field behaviors, using Motion Sequencing (MoSeq) on both male and female mGluR1 KD and control (Ctrl) animals (Fig. 6a). MoSeq platform identifies and quantifies the use of the brief action motifs (e.g., rearing, running, sniffing, referred to as syllables; Supplementary Videos) out of which mouse behavior during exploration is organized (Fig. 6b)[66,67]. We observed that, while gross motor functions (velocity and total distance traveled within a session) were not affected (Supplementary Fig. 6a, b), the exploration patterns of Ctrl and KD mice were distinct (Fig. 6c). We further investigated the behavior motifs involved in this difference with syllable examination and found that no new syllables were used when comparing the mGluR1 KD with control mice, but that the distribution of syllables usage was significantly altered (Fig. 6d). Most of these syllables, such as different types of grooming and rearing, are important parts of mouse exploratory behaviors. Linear discriminant analysis (LDA) clearly separated Ctrl and mGluR1 KD mice, suggesting clear differences in syllable usage (Fig. 6e). We examined the syllables that distinguished Ctrl and KD mice, by sorting the absolute model weights in the LDA and we identified the top ten syllables, in which rearing was highly enriched (7 rears, 2 grooms, and 1 scrunch, Supplementary videos). While the syllable usages were different between Ctrl and KD mice, the velocity during the syllables between the two groups were comparable (Table S5). We repeated LDA on subsets of sessions and further identified two groups of syllables that robustly distinguished Ctrl and KD mice across iterations by sex. The use of these syllables alone could predict whether a given mouse was a Ctrl or a KD mouse well above chance (Fig. 6f). These data demonstrate that deficits in mGluR1 function in SST cINs, which we have shown lead to pervasive developmental phenotypes, are associated with significant modifications of their natural exploratory behaviors, as identified by unsupervised analysis. We verified that these behavioral alterations did not originate from cell death or extensive cortical rewiring, downstream of TC input maturation to SST cINs. While postsynaptic mGluRs have been shown to prevent cell death[68], the postnatal mGluR1 CRISPR-deletion did not induce significant change in the number of infected SST cINs at P30 (Supplementary Fig. 4d). Furthermore, anatomical neural pathways connecting to SST cINs using monosynaptic retrograde rabies tracing were grossly normal, including the adult organization of TC pathways (Supplementary Fig. 7a–g)[69,70]. These results suggest that mGluR1 in SST cINs regulates the proper maturation of adult cortical functions primarily through synaptic refinement, rather than gross anatomical developmental events.

## Discussion

TC afferents to pyramidal and PV neurons become strengthened as development proceeds through Hebbian mechanisms. In contrast, here, we show that early TC connectivity from VB neurons onto SST cINs recruits mGluR1-based transcription, supporting the later elimination of this connectivity known to regulate the development of cortical networks. Moreover, the removal of mGluR1 signaling from SST cINs during this critical period also impacts the development of normal sensory-related exploratory behaviors (Fig. 7).

In electrophysiological development studies, the functional TC inputs to SST cINs during development are referred as transient[13,14,16,71], as they are thought to be absent in adulthood[4,5,72,73]. However, anatomical studies[19,40,74,75] reveal robust TC connectivity to SST cINs in adulthood, suggesting that TC synapses are not completely eliminated after maturation and that remaining synaptic contacts are silent, with respect to fast glutamatergic neurotransmission. Transient connectivity, therefore, only indicates the physiological dynamics of the network, as measured with electrophysiology and as compared to other cell types. Remarkably, synaptic pruning or elimination in

standard Hebbian developmental plasticity, is associated with synaptic strengthening. However, we reveal that transient TC connectivity is associated with a decrease in synaptic contact density. The discrepancy between physiological and morphological synaptic assembly highlights the essential need to better understand these types of processes in a given context, transient connectivity in particular, which would warrant future high-resolution investigations using 3D electron microscopy[76].

Guidance molecules are responsible for circuit wiring and are generally associated with global axon growth during embryonic development[77]. In contrast, postnatal activity-dependent mechanisms of pruning and strengthening are usually linked to neurotransmitter receptors and their composition[23,78,79]. Here, our results reveal a tight interplay between these two mechanisms, wherein metabotropic glutamatergic receptor signaling, rather than cell excitability controls expression levels of guidance molecules. Linking neural activity with genetic programs of synaptic assembly development is essential for a better understanding of neurodevelopmental disorder etiologies. Indeed, a lack of general synaptic refinement has been associated with autism spectrum disorders (ASD)[80,81]. Moreover, disruption of transient brain structures and defects in inhibitory and excitatory balance have been found in autism spectrum disorders (ASD) in particular[82]. However, the role of TC transient connectivity to SST cINs in cortical dysfunctions found in neurodevelopmental disorders remains unexplored.

While mGluR1 is highly specific to SST cINs, primarily during development and in the cortex, mGluR1 expression is also found at high levels in the cerebellum and in the thalamus. Remarkably, previous studies using mGluR1 constitutive KO, demonstrate that mGluR1 promotes synapse elimination, a process leading to synaptic strengthening and synapse maintenance in the cerebellum and the thalamus[47,59,83]. However, we reveal that mGluR1-dependent synapse refinement in SST cINs controls the weakening of presynaptic TC inputs. The comparison of the findings from distinct models reveals that mGluR1 functions are highly contextualized based on both regional and temporal contexts. The CRISPR KD approach thus enabled us to further study behavioral consequences of cell-specific mGluR1 functions, while the constitutive KO mice display strong ataxia and important motor deficits precluding any behavioral study.

SST cINs in L5 of S1 are among the earliest-born populations within this region. Previous works have demonstrated that they play a critical developmental role in regulating the PV cIN maturation and the formation of the feedforward inhibition to pyramidal neurons. These works, in combination with our present findings, illustrate that SST cINs play a key role in regulating the maturation of the S1 cortex that is distinct from those involving Hebbian plastic mechanisms utilized to shape the TC afferents to other cortical neurons. Instead, here, we demonstrate that the dynamic connectivity of TC inputs is regulated through SST cINs using mGluR1-dependent transcriptional regulation, of Sema3A in particular, to directly weaken TC inputs during postnatal development, when the S1 cortex begins to assume its mature function. The discovery of this mechanism illustrates how genetically programmed signals within the SST cIN population allow the thalamic afferents to activate a critical developmental signal that reconfigures S1 cortical circuits. During this period, SST cINs hand over the bottom-up FFI from the thalamus to PV cINs and, in turn, assume their adult role in regulating local feedback signaling to L5 pyramidal neurons. Hence, in contrast to the typical activity-based mechanism by which network dynamics drive cortical maturation, the L5 SST cINs provide a signaling mechanism that initiates the transition from developmental bottom-up signaling to the critical period rebalancing of bottom-up and top-down cortical functions in S1 cortex.

The source of glutamate triggering mGluR1 signaling feedback to TC inputs remains to be investigated. However, we have previously observed, using retrograde labeling, that within glutamatergic inputs

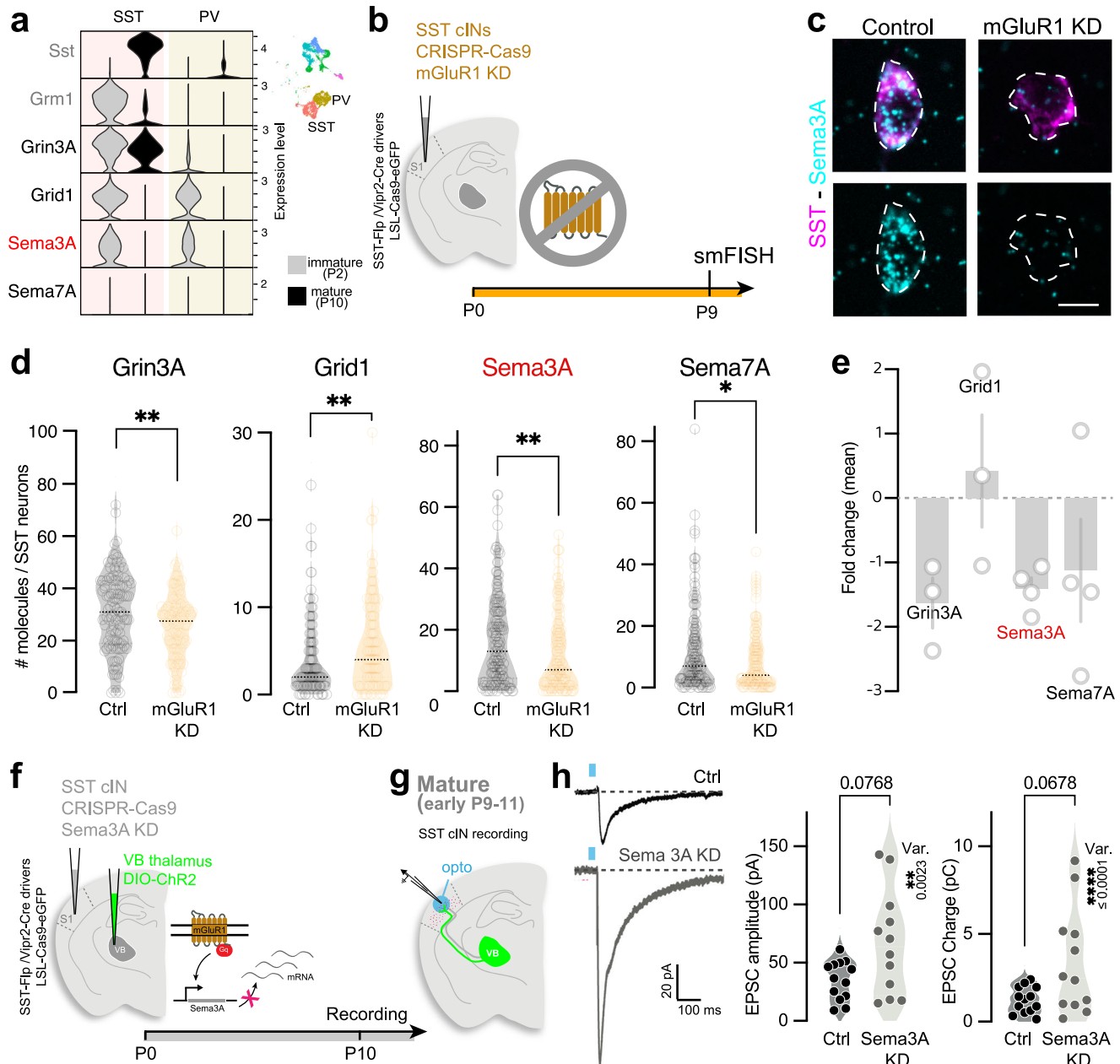

**Fig. 5 | mGluR1-dependent transcriptional regulation in SST cINs regulates transient TC connectivity. a** Early postnatal expression levels of the candidate genes Grin3a, Grid1, Sema3A, and Sema7A in SST cINs using P2/P10 integrated scRNAseq. Violin plots represent the expression of these genes in SST and PV cIN clusters (inset) at P2 and P10, compared to Sst and Grm1. **b** Strategy for investigation of candidate gene expression in SST cINs in the absence of mGluR1. **c** Example of smFISH and SST staining in Ctrl and mGluR1 KD SST cINs at P10. Scale bar: 10 μm. **d** Quantification of RNA molecule per SST cIN of the select candidate genes using smFISH (Grin3a; Ctrl: 30.68 ± 1.44, $n = 111$, $N = 3$; KD: 25.75 ± 1.21, $n = 108$, $N = 3$; two-sided Student $t$-test $p = 0.0096$; Grid1;Ctrl: 4.08 ± 0.39 $n = 124$, $N = 3$, KD: 5.77 ± 0.50 $n = 115$ $N = 3$; two-sided Mann–Whitney test $p = 0.0389$; Sema3A; Ctrl: 16.64 ± 1.21, $n = 148$, $N = 4$; KD 12.04 ± 1.05, $n = 140$, $N = 4$; two-sided Mann–Whitney test $p = 0.0049$; Sema7A; Ctrl 12.04 ± 1.17, $n = 138$, $N = 4$; KD

8.14 ± 0.76, $n = 145$, $N = 4$; two-sided Mann–Whitney test $p = 0.0119$). **e** Fold change per animal (Grin3A: 1.62 ± 0.39, $N = 3$; Grid1: −0.42 ± 0.87, $N = 3$; Sema3A: 1.41 ± 0.17, $N = 4$; Sema7A: 1.12 ± 0.79, $N = 4$). **f** Strategy for early deletion of Sema3A in SST cINs. **g** Recordings of SST cINs upon $TC_{VB}$ stimulation in L5. **h** Averaged EPSCs from unitary examples of Ctrl and Sema3A KD SST cINs at P10. EPSC peak amplitudes and charges in Sema3A KD and Ctrl at P9–11(Amplitude; Ctrl from Fig. 1: 35.08 ± 4.67 $n = 13$, $N = 3$; Sema3A KD: 66.75 ± 12.78 $n = 12$, $N = 3$; two-sided Mann–Whitney test $p = 0.077$; Charge; Ctrl: 1.24 ± 0.21; Sema3A KD: 3.37 ± 0.87; two-sided Mann–Whitney test $p = 0.068$; Bartlett's test, Amplitude: $p = 0.0023$, Charge: $p = 3.15e{-}05$). Data were presented as mean ± SEM. The number of biologically independent animal replicates = $N$ and cell replicates = $n$. Source data are provided as a Source Data file.

received by SST cINs, VB thalamocortical inputs is one of the earliest to connect during development, in contrast to cortico-cortical inputs that increase throughout maturation[19], suggesting mGluR1 activation by TC inputs during early development. Since adult mGluR1 function appears to be distinct from the one during the first postnatal week in S1[44], the timing of glutamatergic input integration could be critical for

triggering feedback signaling from SST cINs. While retrograde labeling does not reveal major disruptions from other inputs onto mGluR1 KD SST cINs, it would be of interest to examine compensatory effects on SST cINs at the level of synaptic physiology.

The feedback mechanism of sensory TC transient connectivity onto SST cINs occurs during critical periods of plasticity, when

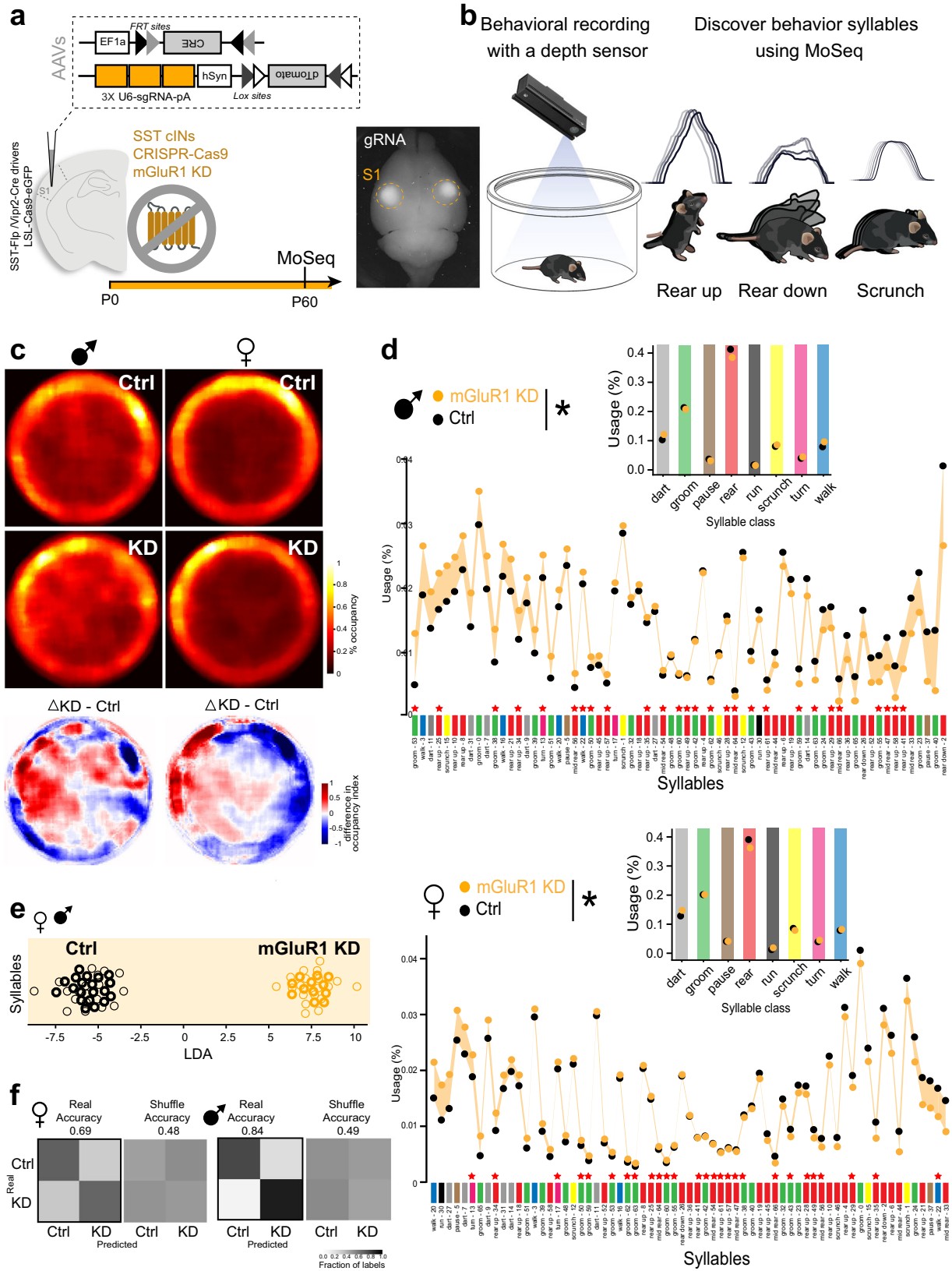

somatosensory topographic maps are highly sensitive to sensory TC inputs[78,84]. In addition, SST cINs have been directly involved in inducing these critical periods of plasticity in the visual cortex[85,86]. Therefore, our findings provide a potential genetically encoded mechanism, by which SST cINs controls TC input-driven cortical plasticity and, ultimately, somatosensory cortical functions. Notably, this mGluR1-based

feedback mechanism is specific to development, as mGluR1 in adulthood has been shown to trigger SST cIN potentiation in adult prefrontal cortex[44]. The opposite effects of mGluR1 at different stages, suggest a distinct SST cIN-specific molecular context, such as Sema3A expression, controlling age-dependent functions. Moreover, this highlights the essential role of these developmental programs, rather

**Fig. 6 | The persistence of developmental TC inputs disrupts exploratory behavior in adulthood. a** Strategy for early deletion of mGluR1 in SST cINs, using CRISPR/Cas9 system (see Fig. 4a). AAVs were injected bilaterally in S1 at P0. Motion Sequencing (MoSeq) was performed around P60 (44 females (24 Ctrl and 20 KD) and 32 males (20 Ctrl and 12 KD) were analyzed). **b** MoSeq pipeline and examples of behavioral motifs (syllables). Modified from ref. 67. **c** Position occupancy heatmap. The top two rows are the min-max scaling of the occupancy percentages (max = 1, min = 0). The bottom row is the difference between occupancy percentages for mGluR1 KD and Ctrl mice (Δ index). **d** Top, Syllables for Ctrl and mGluR1 KD males, sorted by syllable usage differences between mGluR1 KD and Ctrl (Chi-square test on the unnormalized syllable usages *p* = 1e-6). The prominent syllables (see methods) are labeled with red stars (27 syllables for male, and 27 for female). The syllable behavior types are color-coded (i.e., light gray for dart, green for groom, red for

rear, etc.). The inset subpanel above is aggregated usages by syllable classes and the differences between the two groups are significant (Chi-square test on the unnormalized syllable usages *p* = 1e-6). Bottom, Same as d but for female mice (syllable usage differences, *p* = 1e-6; aggregated usages by syllable classes, *p* = 1e-6). **e** Linear discriminant analysis (LDA) plot shows that Ctrl and mGluR1 KD mice are separated by the syllable ensemble (males and females combined). The top ten syllables with the highest absolute model weights consist of 7 rears, 2 grooms, and 1 scrunch. **f** Confusion matrices for classification accuracy of a linear classifier trained on the 19 prominent syllables identified from 76 LDA iterations by sex (see methods). The use of these syllables alone could predict Ctrl or KD mice. The classifier yields to 0.69 and 0.84 accuracy across cross-validation for females and males, respectively (top/bottom). Labels per-mouse are shuffled for control (right). Source data are provided in Zenodo repository[109].

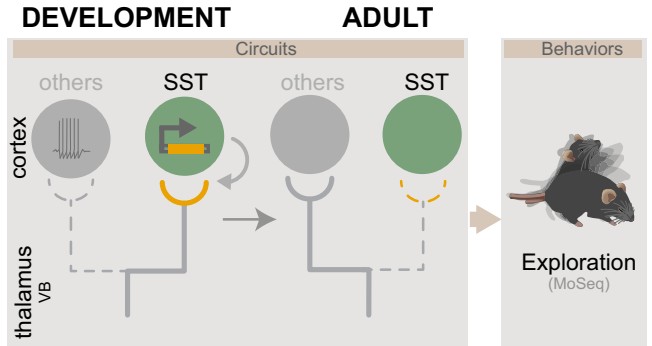

**Fig. 7 | Model for transient TC connectivity development.** In contrast to other neuron types, to which TC connectivity develops according to Hebbian mechanisms, SST cINs receive transient TC inputs from the VB thalamus. These TC inputs regress onto SST cINs following a non-Hebbian mechanism, and they involve postsynaptic metabotropic-dependent transcriptional regulation. This process is required for the maturation of exploratory behaviors in adult mice.

than adult plasticity mechanisms for the formation of proper adult cortical circuits. It remains to be determined whether these developmental genetic programs in SST cINs are sufficient to instruct feedback transient connectivity and input/output circuit reorganization. Interestingly, our postnatal scRNAseq analysis revealed that mGluR1 is also expressed developmentally in VIP cINs (Sncg⁺ and Vip⁺ cIN clusters; Fig. 3a,d), which have been shown to receive transient TC inputs during development[87]. Therefore, mGluR1 is a strong candidate for intrinsic encoding of developmental transient TC connectivity and bottom-up/top-down circuit wiring. The model in which cell type genetic identity is sufficient to determine feedback TC input maturation provides a powerful framework to generate circuit diversity from common TC afferents, such as the formation of feedforward versus feedback inhibition formed by PV and SST cINs respectively.

## Methods
### Mice
All experiments were approved by and in accordance with Harvard Medical School IACUC protocol number IS00001269 and by Cold Spring Harbor Laboratory IACUC protocol number 22-4. Animals were group-housed and maintained under standard, temperature-controlled laboratory conditions. Mice were kept on a 12:12 light/dark cycle and received water and food *ad libitum*. C57Bl/6 mice were used for breeding with transgenic mice. Transgenic mice, SST-Cre (stock number: 013044)[88], SST-FlpO (stock number: 031629)[89], Vipr2-Cre-neo (stock number: 031332; RRID:IMSR_JAX:031332)[90], Calb2-Cre (stock number: 010774; RRID:IMSR_JAX:010774)[88], LSL-Cas9-eGFP (stock number: 026175)[91], RCE:LoxP (stock number: 032037)[92], PV-Cre (stock number: 017320; RRID:IMSR_JAX:017320)[93] are available at Jackson Laboratories. Mice were injected at P0 and experiments

conducted between ages P3–P7 for developmental stages and between ages P9–11 and P28–37 for adult/mature time points. Both female and male animals were used for all experiments.

### Histology
**Immunohistochemistry (IHC).** Mice were deeply anesthetized with sodium pentobarbital by intraperitoneal injection and transcardially perfused with PBS 1X followed by paraformaldehyde (PFA) diluted at 4% in PBS. Brains were postfixed for 4 h in 4% PFA at 4 °C, except for monosynaptic rabies tracing, for which brains were postfixed overnight. 50 μm vibratome (Leica) sections were incubated 1–2 h at room temperature in a blocking solution containing 3% Normal Donkey serum and 0.3% Triton X-100 in PBS and incubated 48 h at 4 °C with primary antibodies: rat anti-RFP (1:1000; Chromotek #5f8), chicken anti-GFP (1:1000; Aves Labs #1020), rabbit anti-Homer1b/c (1:500, Synaptic Systems #160023), guinea-pig anti-VGlut2 (1:2000, Millipore #AB2251), guinea-pig anti-VGlut2 (1:1000, Synaptic Systems #135404), rabbit anti-somatostatin (1:3,000; Peninsula Laboratories International T-4103.0050), and rabbit anti-mGluR1a (1:1000; Af811 Frontier Institute), which exhibits the best labeling compared to multiple anti-mGluR1 we tested (negative controls in constitutive mGluR1 KO did not show any signal). Sections were rinsed three times 15 min with 0.1% Triton X-100 in PBS and incubated for 60–90 min at room temperature or overnight at 4 °C with Alexa Fluor 488-, 594-, or 647-conjugated donkey secondary antibodies (1:500; Thermo Fisher Science or Jackson ImmunoResearch). Sections were mounted in Fluoromount-G (Southern Biotechnology, #100241-874) before imaging.

**Dual single-molecule fluorescent in situ hybridization (smFISH) and immunohistochemistry (IHC).** For combined smFISH and IHC, mice were deeply anesthetized with sodium pentobarbital by intraperitoneal injection and transcardially perfused with PBS followed by 4% PFA. Brains were fixed 1 h at room temperature and 2 h in 4% PFA at 4 °C. The brains were cryopreserved in 30% sucrose before being sectioned at 20 μm using a sliding microtome (Leica). Sections were stored in a cryo-solution containing 28% (w/v) sucrose and 30% (v/v) ethylene glycol in 0.1 M sodium phosphate buffer, pH 7.4, before performing smFISH experiments. RNAscope® fluorescent in situ hybridization and immunohistochemistry (Dual FISH-IHC) assay was performed using RNAscope® Multiplex Fluorescent Detection Kit v2 (ACDBio 323110) purchased from Advanced Cell Diagnostics (ACDBio). The manufacturer's protocol for fixed frozen tissue was followed. Probes used in this study include Mm-Grm1 (ACD 449781), Mm-Sema7A (ACD 437261-C3), Mm-Sema3A (ACD 412961). After smFISH, IHC was performed. Sections were washed in Tris-buffered saline, pH 7.4, (TBS) with 10% Tween®20 (TBST) and blocked in 10% Normal donkey serum in TBS-0.1% bovine serum albumin (BSA) for 1 h at room temperature. Primary antibodies, rat anti-RFP (1:1000; Chromotek #5f8) to reveal AAV-sgRNA-DIO-dTomato reporters, and rabbit anti-somatostatin (1:3000; Peninsula Laboratories International T-4103.0050) diluted in TBS-0.1% BSA, was applied to the sections for

2 h at room temperature. Sections were washed in TBST and incubated in corresponding Alexa-conjugated secondary antibodies (1:500; Thermo Fisher A-10040), in TBS-0.1% BSA, for 1 h at room temperature. Sections were washed in TBST and DAPI (5 uM; Sigma D9542) was applied to each section for 30 s. Prolong Gold antifade mounting medium (Thermo Fisher P36930) was used to adhere the glass coverslip.

## Mouse neonate stereotactic injections

For postnatal time points stereotaxic injections were possible using a neonate adapter (Harvard apparatus). Mouse pups were anesthetized by hypothermia for 5 min and stereotaxically micro-injected with AAV (volume 80–100 nl) at P0-P1 with a micro-injector (Nanoject III). Primary somatosensory cortex (S1) was targeted with the following coordinates: AP + 1.40, ML-1.85, DV-0.20 from Lambda. For slice physiology experiments, double injection of in S1 cortex and the somatosensory thalamus of Vipr2-Cre or Cab2-Cre mice were performed at P0. Thalamus coordinates for VB and PO were as follows: AP + 0.9, ML-1.0, DV-2.4 from Lambda. For more consistent targeting, cortical injections were performed over 5 min and thalamic injections over 10 min, following a 20 nl, 4–5 times with 10 s delay program. After surgery, mice were given Meloxicam (Metacam) subcutaneously at 5 mg/kg of body weight and, upon recovery on a heating pad set at 37 °C, were placed back in the home cage with the mother.

## Chronic DREADD activation

For DREADD experiments, clozapine-N-oxide (CNO, Tocris) or CNO dihydrochloride (Hello Bio) was dissolved in 5% dimethyl sulfoxide (Sigma) or 0.9% saline respectively, at 1 mg/ml (10x solution) and stored at 4 °C for the duration of the chronic activation (7 days). Every day, fresh 1x CNO solution was made by fresh dilution with 0.9% saline to 0.1 mg/ml. Pups were injected with CNO (10 ul/g equivalent to 1 mg/Kg) subcutaneously for 7 days (from P1–P8), twice daily.

## Adult stereotactic injections

For monosynaptic rabies retrograde labeling, stereotactic injection of EnVA-CVS-N2c-dG-H2B-tdTomato (70 nL; dilution 1/10, Addgene plasmid: #175441)[19] was performed at P30 in animals previously injected with AAV-3x gRNA-DIO-TVA-N2cG at P0 (80 nl), using a Nanoject III at 1 nl/s according to stereotaxic coordinates (from Bregma AP + 1.00, ML-3.00, DV-0.89). Animals were perfused 7 days later and processed for immunohistochemistry.

## Viruses

The enhancers, reporters, and effectors were cloned using the Gibson Cloning Assembly Kit (New England BioLabs, catalog no. NEB-E5510S). After cloning and sequencing, the growth time of the transformed DH10B Competent Cells was kept below 12 h on plates and 10 h in flask at 37 °C. DNA from several clones was recovered with an endotoxin-free midi-kit (Zymo D4202). This allows for consistently obtaining low recombination rates detected by PCR. Primers were designed for amplification of plasmids at the junction of each possible recombined form and compared to recombination-free controls. Only clones with a recombined/not recombined ratio of 1.0E + 05 or greater were considered for further AAV production. The rAAVs were produced using standard production methods. Polyethylenimine was used for transfection and OptiPrep gradient (Sigma) was used for viral particle purification. Titer was estimated by quantitative PCR with primers for the WPRE sequence that is common to all constructs.

AAV2/1-hSyn-fDIO-DREADD-Gi-mCherry and hSyn-fDIO-DREADD-Gq-mCherry were produced from gifts from U. Gether (http://n2t.net/addgene:154867/addgene #154867; http://n2t.net/addgene:154868/#154868). Titer: 1.80E + 12 and 1.72E + 12 vg/ml.

AAV2/1-hSyn-DIO-DREADD-Gi-P2A-tagBFP and DREADD-Gq-P2A-tagBFP were generated from VTKS2 backbone[94] and produced for this manuscript. Titer: 3.6E + 12 and 9.1E + 11 vg/ml.

AAV8-hSyn-DIO-HA-hM3D(Gq)-IRES-mCitrine and -HA-hM4D(Gq)-IRES-mCitrine were gifts from B. Roth (http://n2t.net/addgene:50454/addgene #50454-AAV8; http://n2t.net/addgene:50455/addgene #50455-AAV8). Titer: 1.9E + 13 vg/ml.

AAV2/1-3x Grm1 gRNA-DIO-dTomato was generated from the VTKS2 backbone and produced for this manuscript. Titer: 2.42E + 12 vg/ml.

AAV2/1-3x Grm1 gRNA-DIO-N2cG-TVA was generated and produced for this manuscript. Titer: 7.8E + 12 vg/ml.

AAV2/1-3x Grm1 gRNA-fDIO-dTomato was generated from the VTKS3 backbone and produced for this manuscript. Titer: 2.0E + 12 vg/ml.

AAV2/1-3x Sema3A gRNA-DIO-mCherry was generated from the VTKS2 backbone and produced for this manuscript. Titer: 1.07E + 12 vg/ml.

AAV2/9-Ef1a-DIO-ChRmine-mScarlet was produced by the Children's Hospital Viral Core from a plasmid, which was a gift from K. Deisseroth[95] (http://n2t.net/addgene:130998/addgene#130998) produced by. Titer 6.26E + 14 vg/ml.

AAV2/1-DIO-mCherry was generated from the VTKS2 backbone and produced for this manuscript. Titer: 1.4E + 12 vg/ml.

AAV-PHP.eB-EF1a-DIO-hChR2(H134R)-eYFP and AAV9-EF1a-fDIO-eYFP were gifts from K. Deisseroth (http://n2t.net/addgene:20298/addgene #20298-AAV-PHPeB[96]; http://n2t.net/addgene:55641/addgene #55641-AAV9)[97]. Titer: 1.00E + 13 and 2.00E + 13 vg/ml.

AAV2/1-hSyn-fDIO-HA-Kir2.1 was generated from the VTKS3 backbone and produced for this manuscript. Titer: 2.60E + 12 vg/ml.

AAV9-EF1a-fDIO-Cre was a gift from E. Engel & A. Nectow[98] (http://n2t.net/addgene:121675/addgene #121675-AAV9). Titer: 2.50E + 13 vg/ml.

AAV-PHP.eB-S5E2-dTomato was a generous gift from J. Dimidschstein. Titer: 8.3E + 09 vg/ml[99].

EnvA-pseudotyped CVS-N2c(deltaG-H2B:tdTomato) Rabies construct that we previously generated (Addgene #175441)[19] was utilized for monosynaptic retrograde labeling. Titer: 3.7E + 09 U/ml. The nuclear labeling from H2B:tdTomato improves the automated cell detection and overall accuracy. Rabies were generously shared by K. Ritola. Titer: 1.4E + 08 U/ml.

## CRISPR strategy

For proper deletion of Sema3A and Grm1 genes, three guide RNAs (gRNA) were packaged into one single AAV. Each gRNA is under human U6 promoters and with a scaffold gRNA and a polyA sequence at their 3' end. The whole triple gRNA (3x gRNAs) construct was synthesized by GeneScript with an ApaI site at each end, to be directly inserted into a VTKS2 or VTKS3 AAV backbone[94], upstream of the human Synapsin promoter, which controlled a DIO- or a fDIO-reporter (dTomato or N2cG rabies helpers). Each 3x gRNA was packaged into a single AAV1. Grm1 gRNA with spCas9 PAM sequences were designed by cross-validating best ON- and OFF-target scores from Benchling CRISPR gRNA Design Tool, ChopChop[100,101], and CRISPick (previously known as Genetic Perturbation Platform (GPP)). gRNA suggestions were picked in Exons 1 to 3, to increase the probability of full gene deletion. Sequence of selected Grm1 gRNAs are: 5'-CGATGCTTGATATCGTCAAG-3'; 5'-CGACCGCGTCTTCGCCACAA-3' and 5'-GTCGCTCAGGTC-TATGCTCG-3'. Sequence selected for Sema3A gRNAs are: 5'-TACTCCG TTCTTCATCCAGA-3'; 5'-TGTGGCCAGTATCTTACACA-3'; 5'-GAGACG TTAGTGTTGCCATG-3'.

Of note, since each guide have their own off-target effects, it has been increasingly accepted that scrambled gRNAs are not proper controls for CRISPR, in contrast to shRNA or siRNA. In this manuscript, Cas9(+) or animals without gRNA infection were considered as

controls. Alternatively, gRNA-DIO-reporter(+) cells were used as controls. In electrophysiology experiments, the Vipr2-Cre mouse line was utilized for opsin expression in the thalamus. Vipr2-Cre recombination is specific to the first-order thalamus and practically absent from the entire cortex (Allen Brain Atlas Transgenic Characterization)[64], preventing Cas9 recombination in non-SST cINs. SST-Cre alone was not sufficient to trigger Grm1 deletion by P5 and P10, when injected at P0. Instead, we used a combination of Vipr2-Cre and SST-FlpO mouse lines, together with an AAV-fDIO-Cre and the AAV-sgRNAs to trigger CRISPR-deletion in SST cINs early in postnatal development.

## 10x single-cell RNAseq

Publicly available scRNAseq datasets for P2 and P10 cINs were used[36,37] (Gene Expression Omnibus (GEO) at accession number GSE165233 and GSE104156, respectively). P2 scRNAseq was obtained from Dlx6a-Cre; Ai14(+) sorted cINs and prepared with 10x v3.1 Genomics Chromium platform, with a total of 5384 cells sequenced. P10 scRNAseq was obtained from Dlx6a-Cre; RCELoxP sorted cINs and prepared with 10x Genomics Chromium system, with a total of 6346 cells sequenced. While public data are already processed, we aligned the same and higher quality criteria to both datasets, by removing cells with a number of UMI of 500 minimum and 10,000 maximum and cells with higher than 10% mitochondria. The clustering of both datasets was performed using the Seurat pipeline in "R"[38]. The number of principal components used for clustering analysis were determined with the ElbowPlot function (20 dimensions). Cells within P10 and P2 were assigned to cIN type via Seurat canonical correlation analysis (CCA) and clusters were assigned based on gene marker expression.

Cells within non-cIN clusters (excitatory neurons and glial cells) were defined based on the expression of their developmental markers (Slc17a7, Neurod6, Mki67, Mbp, and Gfap)[102] and removed from the UMAP representation.

Marker expression of cIN types were defined as described in ref. 37 for the main cIN types (such as SST and PV cINs) and supported with[40] to assign SST cIN subpopulations (such as Myh8+ and Chodl+ SST cINs). Marker genes for early postnatal PV cINs are Mef2c, Erbb4, Plcxd3, and Grp149; marker genes for early postnatal SST cINs are: Sst, Tspan7, and Satb1. For subpopulations, we verified the expression of adult marker genes in the integrated P2–P10 datasets. SST cIN subtype marker genes expressed at P2/P10 are: Cbln4 (T-shaped Martinotti cINs), Hpse (L4-projecting cINs), Myh8 (T-shaped Martinotti cINs), and nNos1/Chodl (long-range projecting cINs).

Using Seurat, we performed differential gene expression (DGE) between identified SST cIN and PV cIN clusters, between SST cINs and all other cIN clusters, and between SST-Myh8 subtype with all other cIN subtypes, based on the non-parametric Wilcoxon rank-sum test. Log2 (fold change(FC)) and adjusted $p$ values of the top candidates are represented in Tables S3, S4. Adjusted $p$ values smaller than 2.22E − 16 cannot be computed by "R". For visible volcano plot representations, a constant corresponding to one-digit value lower than the last positive decimal of the lowest computer $p$ value, as added to $p$ values = 0. Comparisons were done at each age, P2 and P10, separately.

## Whole-cell slice electrophysiology

For the P6–7 time point, injection of DIO-Chrmine at P0 generated stronger TC input stimulation and improved optogenetic variability from the restricted AAV-expression in such short time windows. From P10 throughout adulthood, DIO-ChR2 was sufficient to evoke more consistent responses. P6–7 mice were anesthetized by hypothermia, followed by decapitation. P9–11 and P28–32 mice were anesthetized using isoflurane, followed by decapitation. Brains were dissected out in ice-cold oxygenated (95% O2 and 5% CO2) sucrose cutting solution containing (in mM): 87 NaCl, 2.5 KCl, 2 MgCl₂, 1 CaCl₂, 1.25 NaH₂PO₄, 26 NaHCO₃, 10 glucose, and 75 sucrose (pH = 7.4). About 300-µm thick coronal slices were cut using a Leica VT1000S vibratome through the

primary somatosensory cortex. Slices recovered in a holding chamber with artificial cerebrospinal fluid (ACSF) containing (in mM) 125 NaCl, 20 Glucose, 2.5 KCl, 1.25 NaH₂PO₄, 26 NaHCO₃, 2 CaCl₂, 1MgCl₂ (pH = 7.4) at 32 °C for 30 min and room temperature for at least 45 min prior to recordings. For recordings, slices were transferred to an upright microscope (Scientifica) with oblique illumination Olympus optics. Cells were visualized using a 60x or a 20x water immersion objective. Slices were perfused with oxygenated ACSF in a recording chamber at room temperature at 1–2 ml/min. Recording electrodes (3–6 MΩ) were pulled from borosilicate glass (1.5 mm OD, Harvard Apparatus) with a horizontal P-1000 Flaming Micropipette Puller (Sutter Instrument). For current-clamp recordings, electrodes were filled with an internal solution containing (in mM): 130 K-Gluconate, 10 KCl, HEPES, 0.2 EGTA, 4 MgATP, 0.3 NaGTP, 5 Phosphocreatine, and 0.4% biocytin, equilibrated with KOH at pH = 7.4. For voltage-clamp recordings, electrodes were filled with an internal solution containing (in mM): 125 Cs-gluconate, 2 CsCl, 10 HEPES, 1 EGTA, 4 MgATP, 0.3Na-GTP,8 Phosphocreatine-Tris, 1 QX-314-Cl, equilibrated with CsOH at pH = 7.4. Recordings were performed using a Multiclamp 700B amplifier (Molecular Devices) and digitized using a Digidata 1550 A and the Clampex 10 program suite (Molecular Devices) or, using an EPC 10 amplifier (HEKA) and the Patchmaster software (HEKA). Voltage-clamp signals were filtered at 3 kHz and recorded with a sampling rate of 10 kHz. Recordings were performed at a holding potential of −70 mV. Cells were only accepted for analysis if the initial series resistance was less than 40 MΩ and did not change by more than 20% during the recording period. The series resistance was compensated at least -50% in voltage-clamp mode, and no corrections were made for the liquid junction potential. Whole-cell patch clamp recordings were done from reporter-expressing SST cINs and nearby non-fluorescent pyramidal neurons in L5. fDIO-mCherry was used for developmental analysis and all control conditions throughout the study. To stimulate thalamic afferents expressing ChR2, blue light (5.46–6.94 mW from the objective) was transmitted from a collimated LED (Mightex) to the epifluorescence port of the microscope. We used a 5 ms pulse of light for the stimulation of P7 and P10 slices and a 1 ms pulse of light for P30 slices. Pulses were delivered once every 5 s, for a total of 15 trials. To isolate monosynaptic EPSCs from the thalamus to the cortex, recordings were performed in the presence of 1 µM TTX and 1 mM 4-AP (Tocris). Data analysis was performed using MATLAB and Prism 8 (Graphpad). The peak amplitude of the evoked EPSCs were averaged across the 15 trials per cell.

For intrinsic properties, SST cINs were recorded in current-clamp mode and the data were acquired at 20 KHz. In supplementary Table 1, the resting membrane potential (RMP) of the cell is the potential when the net injected current (holding current + IV current-clamp step) is the closest to zero. Input resistance (IR) was computed using Ohm's law (V = I/R) by finding the slope of the IV curve obtained using the net current from the IV current-clamp protocol sweep with no action potential (AP) and the potential from the corresponding sweeps. Rheobase was defined as the minimum net injected current to evoke a sweep with more than one AP. AP threshold was defined as the max of the second derivative of the first AP from the first sweep with more than one AP. In supplementary Table 2, the resting membrane potential (RMP) of the cell was recorded as mean membrane potential from a 1-min-long recording in I = 0 mode or by taking the mean potential. Input resistance (IR) was computed using Ohm's law (V = I/R) by finding the slope of the IV curve obtained using current injection from −10 to 10 pA in steps of 5 pA. Rheobase was defined as the minimum input current to evoke firing from a cell. The spike threshold was defined as the membrane potential where dv/dt >5 mV/ms before spike initiation. mGluR1 KD intrinsic properties were compared to publicly available Control data of SST cINs in somatosensory cortex at P10, recorded the same way[19]. Analyses were done using clampfit, Easy Electrophysiology, and Python.

Statistical analyses were done using Python (scipy)[103] and in Graph-Pad prism 9.

## Motion sequencing (MoSeq)

SST-Cre:Cas9fl mice were injected with AAVs expressing mGluR1-gRNA, fDIO-Cre bilaterally in the somatosensory cortex at P0-P1. Control animals were SST-Cre:Cas9fl injected with fDIO-Cre-AAV or gRNA-AAV only, or SST-Cre animals injected with both fDIO-Cre- and gRNA-AAVs. All mice were perfused after recording and analysis to confirm the bilateral targeting of the somatosensory cortices, as in Fig. 5b. Mice were recorded around 2.5 months of age.

MoSeq recordings were carried out as previously described[66,67]. 3D depth video is recorded in a circular 17" diameter open-filed arena (OFA) using a Microsoft Kinect v2 depth camera. Briefly, mice were freely moving in the arena for 45 min in the dark, and put back to their home cage. The arena is wiped down with 10% bleach, followed by 70% ethanol between each session. After removing corrupted sessions due to a significant amount of noise in the recording, in total 44 female mice (24 Ctrl and 20 KD) and 32 male mice (20 Ctrl and 12 KD) were used in the analysis.

MoSeq is an unsupervised machine-learning algorithm that segments animal behavior in a novel, but neutral environment (bucket) into repeated modulated motifs called syllables. The set of syllables represents the grammar expressed in the exploration (body language) of each animal and each group. Each syllable is uniquely identified with a number and a global behavior label, such as "rearing" or "grooming", which are identified with a color label in the figures and grouped into global classes (corresponding colors).

MoSeq pipeline (http:www.moseq4all.org) takes 3D depth videos as input and produces frame-by-frame syllable labels for each frame in the videos. Depth videos first go through image processing in the MoSeq package suite to crop out and align the mouse such that the center of the mouse is at the center of the frame, and the nose of the mouse is pointing right. To accommodate noise and morphological variations, the processed aligned video goes through a deep neural network for denoising. The denoised data goes through dimensionality reduction and modeling steps to generate the syllable labels. Syllables are labeled by sorted usage across all sessions such that syllable 0 is the most used syllable, and the syllables that makeup 99% of the total frames are included in the analysis. Syllable usages by group are aggregated across all sessions in the group (i.e., male Ctrl, male KD, female Ctrl, and female KD) and normalized such that syllable usages within one session add up to 1. Chi-square tests are done on the unnormalized frame counts aggregated by group. MoSeq pipeline outputs frame-by-frame kinematic values such as velocity and the coordinates for the mouse centroid in addition to the syllable labels. The position occupancy heatmap for each session is computed by binning the mouse centroid with 100 bins for each axis to compute a 2D histogram, and the histogram is smoothed by a median filter with a kernel size of 11 pixels. The group position density is the average of all the session position densities within the group. The heatmap differences are computed by subtracting the control histogram from the KD histogram for each sex. The average velocity in a session is the average of the frame-by-frame velocity. The total distance traveled within a session is the sum of between-frame mouse centroid movement.

## Linear discriminant analysis (LDA).

To find the syllables that distinguished Ctrl from KD animals, we used LDA on syllable usages across all sessions to find the linear projection that maximizes group separability between Ctrl and KD mice, and sessions from both male and female data in each genotype are combined.

To find a set of syllables that robustly distinguish Ctrl and KD mice by sex, we applied a leave-one-out strategy and trained multiple LDA embedding on a subset of the sessions within the same sex, leaving one session out. The top ten high absolute loading syllables are recorded at each iteration, and the most frequent 10 syllables among all the iterations by sex are used in the linear classifier. There were a total of 27 unique syllables found in the iterations for each sex. The 19 syllables that were present in males and females were used in the classifier.

## Linear classifier on syllable usages.

We applied a linear classifier to the usages of the 19 syllables identified in the leave-one-out LDA described above. Since the number of Ctrl and KD mice included in both males and females are unbalanced, the data is randomly subsampled such that both Ctrl and KD have the same number of mice. The subsample process is repeated ten times. After subsampling, the input data goes through a standard scaler for data preprocessing, and logistic regression is used to predict genotypes. Leave-one-out cross-validation is used to compute model accuracy.

## Quantification and statistical analysis

All statistical details can be found both in the results and the legends of the corresponding figure. In the manuscript, "*N*" represents the number of animals, while "*n*" represents the number of cells. All statistical analyses were performed using GraphPad Prism, Python (scipy), and R software. Unless otherwise tested, statistical significance was tested with one-way ANOVA, followed by post hoc Tukey's multiple comparison test or Student $t$-test for parametric data, and with one-way Kruskal–Wallis followed by post hoc Dunn's test or Mann–Whitney test for non-parametric data. $P$ values $<0.05$ were considered statistically significant. For all figures, $*p < 0.05$; $**p < 0.01$; $***p < 0.001$; $****p < 0.0001$.

The normal distribution of the synaptic density was formally tested with a Kolmogorov–Smirnov test. The data distribution of the synaptic density during development, in CRISPR KD, in Kir2.1, and in DREADD experiments did not show a normal distribution, as well as all smFISH RNA candidates, except Grin3a and they were tested with non-parametric tests (Kruskal–Wallis and Mann–Whitney). The distribution of the other data were normal and tested with parametric tests (Student $t$-test and one-way ANOVA). The data distribution of cell density and monosynaptic retrograde labeling was assumed to be normal and analyzed with parametric tests, based on previous studies, but was not formally tested.

RNA particle distribution (from smFISH) appears to be marker-specific, as previous studies use parametric and non-parametric tests. We formally tested the data distribution of Grm1 RNA particles, found they are primarily normal (with D'Agostino Pearson test. The exception was P10 data, when tested with a Kolmogorov–Smirnov test only, $p$ value = 0.0132) and used parametric tests. In contrast, Sema3A smFISH data distribution was not normal (Kolmogorov–Smirnov test $p$ value $<0.0001$) and non-parametric tests were performed.

The data distribution of optogenetics responses was assumed to be non-gaussian[4,104,105] and analyzed with non-parametric tests, based on previous studies, but was not formally tested.

## Image acquisition and analysis

For all analyses, images were taken in layer 5 of the somatosensory cortex, except for cell number quantification, which was performed in all layers. For analysis of synapse density and smFISH, tissue samples were imaged on an upright ZEISS LSM 800 confocal using a 40X oil immersion objective, 1.4 NA, 2.5 digital zoom, 2048 × 2048 pixels, 16 bits. For cell density and monosynaptic retrograde labeling, tissue samples were imaged on a ZEISS Axio Imager using a 10X dry objective (with tiling mode).

## Synaptic density.

For TC synapse analysis onto PV and SST cINs, single planes were analyzed using a custom script in Fiji (ImageJ) software, as previously described before[102]. In brief, noise reduction and smoothing were applied to all channels and the images were converted to RGB. A color threshold was automatically set to identify the cell body of SST cINs and PV cINs labeled with the RCE:LoxP mouse line. In contrast, the

reporter distribution of DREADD-Gi/Gq and as well as the somatostatin labeling were not homogeneous. In that case, manual delimitation of cell bodies was performed. Cell body perimeter was automatically measured and a masked binary image with the cell body only was created. For bouton segmentation, a watershed-based method is used such that boutons were separated based on the local minima of the pixel gray values. For the presynaptic boutons (VGluT2) and postsynaptic clusters (Homer1), a color threshold was selected to segment boutons as isolated puncta. The comparison between the original images and the masks was used to guide the choice of the threshold value, which was determined to detect all putative presynaptic boutons or postsynaptic clusters in each condition. The "Analyze Particles" (VGluT2 minimum size = 0.20 mm; Homer1 minimum size = 0.10 mm) and "Watershed" tools were applied and a mask was generated. A merged image from all masks was created, converted to an 8-bit image and the overlap between presynaptic TC puncta, postsynaptic clusters and the cell body was automatically detected.

**smFISH analysis.** Single RNA molecules were quantified within SST cINs, as labeled with somatostatin antibody. Control cINs were imaged on the non-injected contralateral side, while KD cINs were imaged on the injected ipsilateral side and identified by the AAV-reporter colocalization. Single plane images were analyzed using a similar adapted custom script in Fiji (ImageJ) software. Images were converted in RGB, and cell bodies were first identified as control or KD and then manually delimited based on the somatostatin labeling. Grm1 RNA, in particular, formed dense clusters within the cells and the watershed-based method was used to separate particles. A color threshold was applied to all images of the same condition to segment RNA particles. The "Analyze Particles" (minimum size for Grm1 was determined as -0.30 mm) and "Watershed" tools were applied and a mask was generated. A merged image of the two masks was created, converted to an 8-bit image, and the overlap between SST cINs and RNA dots automatically detected.

**Cell density.** For SST cIN density in DREADD experiments at P10, SST cINs were labeled with the AAV-DIO-reporter, mCitrine for DREADD-Gi/Gq. SST cIN infected with AAV-DIO-DREADD-Gi-mCitrine, from mice injected with saline instead of CNO, were the control condition. For SST cIN density in mGluR1 KD CRISPR experiments at P30, SST INs were identified from the LSL-Cas9eGFP labeling. Control cINs were infected by fDIO-Cre only without AAV-sgRNA. Using Fiji (ImageJ) software, areas covering ~3–4 topographical columns of the somatosensory cortex were drawn from layer 1 to the white matter. Area (mm$^2$) was measured to normalized cell number and SST cINs were manually counted using the "cell counting" plugin. Quantification of one to three sections per DREADD(+) and four sections per CRISPR KD and Ctrl brains were averaged to represent one biological N each. The identification of each condition was blind to the experimenter.

**Monosynaptic retrograde labeling quantification.** Every fourth section of a whole brain (from orbital cortex to brainstem) was imaged and uploaded into NeuroInfo® software (MBF Bioscience). All sections were manually reordered from the rostral to the caudal direction of the brain. The software's section detection parameters were adjusted to properly recognize the borders of each brain section. Sections were aligned, first using the software's "Most Accurate" alignment option, and then re-adjusted manually if necessary. The distance between each section (200 μm) was specified for the sections to be registered to a standardized 3D mouse brain atlas, using the "Section Registration" function of the software. For P30 mouse brains, the last version of the Mouse Allen Brain atlas was selected. Non-linear registration was run on each section to account for the slight distortions, such as imperfections from sectioning/mounting and/or asymmetry from the sectioning angle. In the "Cell Detection" function, parameters for cell size

and distance from the background were adjusted, although the use of H2B:tdTomato rabies, which are nuclear, greatly improves the detection specificity compared to other labeling. Neural Network with "pyramidal-p64-c1-v15.pbx" preset was used to automatically detect rabies-infected nuclei in the red channel. Detection results were reviewed manually to correct for any detection mistakes (false positives or negatives). The final results were exported as a CSV file. Retrogradely labeled neurons were grouped into functional regions and normalized by the total of retrogradely labeled neurons per brain. Depending on the analysis, functional groups or subgroups of regions were defined as follows: local = S1 cortex respectively; contralateral = any neurons found in the non-injected contralateral side; other areas = all ipsilateral cortical areas except S1; primary = primary sensory areas: M1, S1, V1, A1; association = "other areas" - "primary areas"; First-order thalamic nuclei: dLG, VB, LD, AV, VAL, MGB; Higher order = LP, PO, VM, MD, AM; Limbic = Pf, AD, CM; CL; PCN; Rhe/RH. Hierarchical functional organization of the thalamus, as first, higher order and Limbic populations reflect the genetic and functional identity of thalamic nuclei[106–108]. This organization of rabies retrogradely labeled thalamic neurons was previously used to identify thalamo-cortical pathway changes after sensory deprivation[19] and help us to distinguish whole thalamus versus subpopulation changes in the function of thalamocortical pathways.

### Reporting summary
Further information on research design is available in the Nature Portfolio Reporting Summary linked to this article.

## Data availability
All histology and electrophysiology data generated in this study are provided in the Supplementary Information/Source Data files. Publicly available scRNAseq datasets for P2 and P10 cINs utilized in this study are available on Gene Expression Omnibus (GEO) under accession numbers GSE165233 and GSE104156, respectively. The data generated for Motion Sequencing have been deposited in the Zenodo repository https://doi.org/10.5281/zenodo.11074522. Source data are provided with this paper.

## Code availability
The Seurat package, which was used for the integration of public developmental datasets and differential gene expression, is open-source and freely available on GitHub (https://github.com/satijalab/seurat) and CRAN. Codes used to analyze the Motion Sequencing data presented in this manuscript are available on GitHub (https://github.com/Pouchelon-Lab/metabotropic_signaling) and provided with the data on Zenodo repository https://doi.org/10.5281/zenodo.11074522.

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

## Acknowledgements

We would like to thank Marian Fernandez-Otero for the extensive support with mouse colony/genotyping at Harvard Medical School and Elaine Sevier for the discussions and none-included preliminary analysis. This work was supported by an EMBO Long-Term fellowship, an early- and advanced Swiss Foundation postdoctoral fellowship, a Hearst Foundation grant and the Pershing Square Innovation Fund (PSIF) (to G.P.), and grants from the National Institutes of Health (NIH), MH071679, NS08297, NS074972, and MH095147, as well as support from the Simons Foundation (SFARI) (to G.F.).

## Author contributions

G.P. and G.F. conceived the project, developed the methodology, and wrote the manuscript. G.P., D.Dwivedi, and D.Dumontier analyzed and interpreted the results. D.Dumontier and G.P. processed experimental data and designed the figures. G.P. designed the CRISPR and viral strategies/constructs. D.Dwivedi performed electrophysiological experiments and analysis. D.Dumontier recorded intrinsic properties from DREADD experiments and performed the corresponding analysis. G.P. analyzed scRNAseq datasets. A.M.C.M. and M.S. established and performed smFISH, respectively. S.Liebman and D.Dumontier performed DREADD-dependent cell death experiments. D.J. quantified cell density and was blind to the experiments. G.P., D.Dwivedi, S.Liebman., M.S., and D.Dumontier performed stereotactic injections. G.P. performed the synaptic density experiments and analysis. Q.X. cloned and produced AAVs. Y.Q. completed the automated monosynaptic retrograde labeling analysis. M.S., G.P., and Y.Q. collected MoSeq recordings. S. Lin. and S.D. processed, analyzed, and interpreted the MoSeq behavioral experiments. S.D. helped write the manuscript.

## Competing interests

Gord Fishell is a founder of Regel Therapeutics, which has no competing interests with the present manuscript. The remaining authors declare no competing interests.
