## [Peer Review File · Nature Communications]

Metabotropic signaling within somatostatin interneurons controls thalamocortical inputs during developmentREVIEWER COMMENTS

Reviewer #1 (Remarks to the Author):

In the manuscript by Dwivedi et al the authors explore the mechanisms by which TC (thalamocortical) inputs weaken onto SST cINs (somatostatin cortical interneurons) and the molecular players that mediate this process. The authors found that the transient connectivity is inversely correlated to neuronal activity, mediated by the (metabotropic) mGluR1 receptor in SST cINs, by the transcriptional regulation of downstream gene transcriptional regulation of semaphorin 3A (Sema 3A). They also found that postsynaptic SST cINs actively regulate circuit development which ultimately underlies the proper function of exploratory behaviors in adult mice.

The manuscript follows a coherent experimental progression and offers intriguing mechanistic insights. Nevertheless, I have various points that require clarification and discussion with the authors:

1. One aspect that appears unclear to me is the variation in ages used in different experiments. This leaves me with the impression that there might be a "missing" correlation between the measured activity and the extent of thalamocortical (TC) inputs. For example, in Figure 1, the authors demonstrate that TC input from the ventro-basal (VB) region transiently projects onto SST cINs during postnatal development (measuring inward currents in SST cINs at P6/10/30). The recording details in Figure 1C (left) are not explicitly stated if they represent mean values. Nonetheless, the decrease in current is apparent, particularly in terms of current amplitude. However, the distinction between P6 and P7 is not as clear, especially when examining the number of synapses per cell (refer to Figure 1D comparing P7 vs P30). Are P6 and P7 considered different in this context?
2. Next, the authors observe that the postsynaptic activity of SST cINs influences the maturation of transient thalamocortical (TC) inputs. Through DREADD-Gq activation, they find that this promotes the expected weakening of synapses, not the anticipated activation of Gi. Additionally, they verify the lack of effect of Kir2.1 expression on TCvb neurons, noting no impact. On average, these experiments suggest an inverse correlation between TC transient connectivity and postsynaptic activity of SST cINs during postnatal development. However, there is an issue with this figure as the traces depicted in Figure 2c (for current) do not align with the values shown in Figure 2d. Based on the traces, the Gi amplitude should be greater than the control amplitude, but this discrepancy is not reflected in the quantification. Consequently, the conclusion drawn from this figure is not entirely clear.
3. In the subsequent figure, the authors investigate the metabotropic Gq-signaling that might underlie the attenuation of TC afferents to SST. They analyze public single-cell RNAseq data at P2 and P10. However, the connection between these stages and their previous findings (Figures 1 and 2) is not clear. Despite this, the data suggest that mGluR1 could be the candidate responsible for initiating metabotropic postsynaptic signaling in SST to regulate transient TC inputs during early postnatal development. However, the rationale behind selecting four transcription factors (TF) from the list provided in Supplementary Table 2 (Grin3A, Grid1, and two guidance molecules Semaphorin 3A and 7A) is not apparent. Could the authors provide clarification on this choice?
4. In their final figure (Figure 6), the authors examine the sequences in the Motor Sequencing platform in control and mGluR1 KD mice. They identify a set of syllables (10 syllables) that are highly enriched in the knockout (KO) animals. However, it is unclear to me how to establish a consistent link between the alteration in Sema3A's effect on axon guidance and the set of syllables increased in KD mice. Could the authors provide an explanation or clarify the connection between these findings?

In summary, the manuscript poses challenges in integrating information, with a notable lack of correlation between exemplar experiments and overall values being a prominent example. The use of different ages in various experiments raises concerns about the correlation between measured activity and the degree of TC inputs. The age-related comparisons, especially in Figure 1, lack clarity.

Additionally, the discrepancy between traces in Figure 2C and the quantification in Figure 2D raises questions about the conclusions drawn from this figure. The rationale behind selecting specific transcription factors (TF) in the analysis of mGluR1 signaling and the connection between alterations in Sema3A's effect on axon guidance and the syllable sequences in KD mice need further explanation. Overall, integration of information throughout the manuscript appears challenging, warranting a more coherent narrative. Further the relevance of Figure 7 is also needed (it is not commented in the text).

Reviewer #2 (Remarks to the Author):

This study brings novel insights into molecular mechanisms underlying synaptic remodeling of cortical circuits during perinatal development. Synaptic pruning is clearly a very important developmental process with mounting evidence that it is disturbed in developmental disorders, thus identifying new mechanisms is of importance

Here the focus is on the early connection linking the thalamus to cortical somatostatin interneurons. The Fishell laboratory had previously shown that SST neurons are the first target of thalamocortical axons (TCAs) and that this connection is greatly reduced during development, with TCAs shifting their connection to parvalbumin interneurons. Using chemogenetic approach that targets selectively the SST neurons they find that the pruning of this exuberant TCA connection is increased by the activation of excitatory DREADDs whereas inhibitory DREADDs or Ki67 have little effect. They identify the metabotropic mGluR1 receptor as a potential candidate for this: downregulation of mGluR1 in the SST neurons prevents the normal pruning of the TCA-SST connectivity. Interestingly, knockdown of mGluR1 in SST neurons downregulates the expression of a number of transcripts including a number of repellent guidance molecules such as Sema3a and sema7a which could participate in the synaptic remodeling. Finally they report that inactivation of mGluR1 in SST cells has some effects on mouse behavior that they interpret as a disruption in exploratory behavior.

The experiments are state of the art combining a number of complementary cellular, electrophysiological and molecular approaches with appropriate controls. The paper is beautifully illustrated. My comments are minor and concern some interpretations that could be reconsidered and text clarifications.

1) The specific targeting of SST neurons in all the experiments is particularly important for the clear demonstration of a role of mGluR1 in postsynaptic cells in the pruning of presynaptic afferents. In this regard it would be important that the authors place their finding in the perspective of finding in mglur1-KO mice that had indicated a somewhat similar role of mGluR1 in synapse elimination in other neural circuits such as the elimination of climbing fibers in the cerebellum (e.g. Kano et al.1995), the lemniscal input to VB neurons (10.1371/journal.pone.0226820).

2) In much of the text, the authors refer to the TCA-SST connection as being transient. However from the data presented in the present study, and previous papers of the team, and other studies, the TCA-SST connection is still present in mature circuits. So this is more a remodeling process than a complete elimination of a connection, or a transient connection. Terminology should be adapted to match better this fact.

3) To evaluate synaptic connectivity the study relies largely on morphological analyses that do not have the necessary resolution to qualify synapses. Although Vglut2/Homer1 is a convenient proxy to estimate synaptic input close to a neuron, these are still synapse-like boutons, as stated in the material and method section, which could be stated upfront in the result section. In the material and method section please indicate the "thickness" of confocal plane analysed.

4) The significance of changes in behaviour are difficult to understand and even more to interpret. The same behaviours (e.g. "rear up") seem alternatively as increased or decreased in the KD animals (figure 6c). It seems therefore difficult to qualify these as disruption in exploration. More standard tests of exploration should be used to substantiate this interpretation .

5) The discussion could be streamlined on the main topic of synaptic remodeling, and interaction of activity with guidance molecules in this process.

Reviewer #3 (Remarks to the Author):

In the present study, the authors characterized transient thalamocortical inputs in the developing mouse sensory cortex. They confirmed that thalamocortical inputs from the VB transiently project to somatostatin (SST) inhibitory cortical interneurons (cINs) during early postnatal development. They found that enhancing the postsynaptic activity of SST cINs employing DREADD-Gq promoted the weakening of the TC inputs to SST cINs whereas reducing the postsynaptic activity by DREADD-Gi or by expressing Kir2.1 in SST cINs had almost no effects. They showed that the type 1 metabotropic glutamate receptor (mGluR1) was highly expressed in SST-cINs during development and found that CRISPR-mediated deletion of mGluR1 from SST-cINs prevented the weakening of the transient TC inputs to SST-cINs. Among candidate genes that are regulated by mGluR1 in SST-cINs based on their single-cell RNA sequencing analyses, they identified Semaphorin 3A as a molecule that mediates the effect of mGluR1 to weaken the transient TC inputs to SST-Ins. Moreover, by using the Motion Sequencing platform, they showed that mice with mGluR1 deletion in SST-cINs exhibited impairment of parameters related to exploratory behaviors. The authors argue that mGluR1 in SST-cINs regulates the weakening of TC inputs to these neurons during early postnatal development and this developmental synaptic refinement is required for the development of exploratory behavior in adult mice.

The authors used several state-of-the-art techniques to draw their conclusions and the data appear convincing. The results are very interesting and may attract readers of various neuroscience fields. I have several comments and requests as listed below.

Major Comments

1. There is a clear discrepancy between the electrophysiological data for EPSCs (Figure 2d) and morphological data for synapse density (Figure 2e). While DREADD-Gi did not affect the EPSC amplitude and charge compared to control (Figure 2d), it significantly increased the synapse density (Figure 2e). Moreover, while DREADD-Gq markedly reduced the EPSC amplitude and charge compared to control (Figure 2d), it did not affect the synapse density (Figure 2e). It seems possible that silent synapses with no postsynaptic AMPA receptors may increase by DREADD-Gi and that functional synapses may become silent while keeping synaptic structures by DREADD-Gq. To check this possibility, the authors may try immunostaining of AMPA receptors and check whether the density of VGluT2 puncta that does not colocalize with AMPA receptor signal changed after DREADD-Gq and DREADD-Gi.

2. Throughout the manuscript, the authors describe that the transient connectivity from the VB to SST-cINs inversely correlated with "postsynaptic neuron activity." This argument is mainly based on the results of the DREADD-Gq experiment (Figure 2, Supplementary Figure 2). However, the activation of DREADD-Gq induces not only depolarization leading to action potential generation but also driving Gq signaling cascade that may not accompany membrane depolarization. Since mGluR1 mediates the developmental weakening of TC inputs to SST-cINs, the activation of DREADD-Gq is considered to bypass mGluR1 and directly activate the Gq-signaling cascade downstream of mGluR1. It is therefore

unclear whether the depolarization of SST-cINs promotes the weakening of TC inputs. The failure to induce the reduction of the EPSC amplitude and charge by the Kir2.1 expression (Supplementary Figure 2f, g) suggests that the membrane potential of postsynaptic SST-cINs does not play a major role. Since continuous optogenetic manipulation of SST-cINs in developing mouse pups in vivo is difficult, the authors may check whether the DREADD-Gq-induced promotion of synaptic weakening is prevented by the expression of Kir2.1 in SST-cINs.

3. Supplementary Figure 2f, g: The authors check whether the synapse density increases in Kir2.1-expressing SST-cINs similarly to SST-cINs in which DREADD-Gi was activated (Figure 2e).

4. Figure 5f-h: The authors should examine whether the synapse density is increased in Sema 3A-KD SST-cINs as they examined in mGluR1-KD SST-cINs.

Minor points

1. Page 6, lines 118-120 "After CNO discontinuation (P8), these effects proved temporary, as TCVB inputs were not reduced onto Gq(+) SST cINs anymore by P30 (Supplementary Fig. 2e)." The EPSC amplitude of Gq at P30 (57.38 ± 11.75 , $n = 17$, Supplementary Fig. 2e) seems significantly larger than Gq at P10 (12.68 ± 2.021 , $n = 8$, $N = 3$, Fig. 2d). Does this mean that TC inputs to SST-cINs increase after the termination of DREADD-Gq activation while the TC inputs decreased during the same developmental period in control mice? Please comment on this point.

2. Figure 2c: The sample EPSC trace of Gi in Fig. 2c is not representative since DREADD-Gi activation did not change the EPSC amplitude and charge (Fig. 2d).

3. In Figure 6, the data for mGluR1-KD mice are highlighted in bright yellow, but it is rather difficult to see. Furthermore, the letters in Fig. 6c are too small to read. Please use a different color and make the letters larger.

4. Supplementary Fig.2g: In the legend for Supplementary Fig.2g, the values for "Ctrl" are not shown.

Reviewer #1 (Remarks to the Author):

In the manuscript by Dwivedi et al the authors explore the mechanisms by which TC (thalamocortical) inputs weaken onto SST cINs (somatostatin cortical interneurons) and the molecular players that mediate this process. The authors found that the transient connectivity is inversely correlated to neuronal activity, mediated by the (metabotropic) mGluR1 receptor in SST cINs, by the transcriptional regulation of downstream gene transcriptional regulation of semaphorin 3A (Sema 3A). They also found that postsynaptic SST cINs actively regulate circuit development which ultimately underlies the proper function of exploratory behaviors in adult mice.

The manuscript follows a coherent experimental progression and offers intriguing mechanistic insights. Nevertheless, I have various points that require clarification and discussion with the authors:

1. One aspect that appears unclear to me is the variation in ages used in different experiments. This leaves me with the impression that there might be a "missing" correlation between the measured activity and the extent of thalamocortical (TC) inputs. For example, in Figure 1, the authors demonstrate that TC input from the ventro-basal (VB) region transiently projects onto SST cINs during postnatal development (measuring inward currents in SST cINs at P6/10/30). The recording details in Figure 1C (left) are not explicitly stated if they represent mean values. Nonetheless, the decrease in current is apparent, particularly in terms of current amplitude. However, the distinction between P6 and P7 is not as clear, especially when examining the number of synapses per cell (refer to Figure 1D comparing P7 vs P30). Are P6 and P7 considered different in this context?

We have grouped or separated time points, based on specific technique constraints and originally provided the description in the method section. We now clarified that we examined discrete time windows instead of precise time points when using electrophysiology, within the figures (Fig. 1c, Supp. Fig. 1a-c, see example screenshots and all other Fig.), in the legends and within the text (Lines 82-87 and as highlighted throughout the manuscript). Based on our experimental results, we included an "immature" time window (until P7, when no physiological weakening has occurred) and two "mature" windows: an early mature stage (P9 to P11 - By P8, preliminary data (not shown) revealed that SST neuron responses were already strikingly weakened) and a later mature stage (P28-P32). As previously described (Tuncdemir et al., 2016), there is a sharp decrease of synaptic strength right after the *immature* window. The traces shown in Fig. 1c, as well as in the rest of the paper, are averaged traces from unitary example neurons (P6 and P30). This is now clarified in the legend.

The immunohistochemistry of synaptic contacts enables us to study specific time points, instead of windows. For clarity, we added the time window to which each time point belongs (Fig. 1, 2, 4 & 5). Nonetheless, and as pointed out by Reviewer #2, synaptic density is not always linearly coupled with input strength, and the dynamics of synaptic contact refinement shows a sharper decrease as early as P7. This point is now discussed in the manuscript and commented in response to Reviewer #2.

2. Next, the authors observe that the postsynaptic activity of SST cINS influences the maturation of transient thalamocortical (TC) inputs. Through DREADD-Gq activation, they find that this promotes the expected weakening of synapses, not the anticipated activation of Gi. Additionally, they verify the lack of effect of Kir2.1 expression on TCvb neurons, noting no impact. On average, these experiments suggest an inverse correlation between TC transient connectivity and postsynaptic activity of SST cINs during postnatal development. However, there is an issue with this figure as the traces depicted in Figure 2c (for current) do not align with the values shown in Figure 2d. Based on the traces, the Gi amplitude should be greater than the control amplitude, but this discrepancy is not reflected in the quantification. Consequently, the conclusion drawn from this figure is not entirely clear.

We thank the Reviewer for pointing out our suboptimal choice for example traces in Fig. 2d. We have now picked other examples from the same dataset and replaced the example traces in Fig. 2c. Screenshot of the updated panel:

3. For clarity we divide the following comment into subqueries:

A) In the subsequent figure, the authors investigate the metabotropic Gq-signaling that might underlie the attenuation of TC afferents to SST. They analyze public single-cell RNAseq data at P2 and P10. However, the connection between these stages and their previous findings (Figures 1 and 2) is not clear.

We examined SST cIN-specific gene expression during the formation and maturation of TC transient connectivity. Publicly available scRNAseq datasets for enriched, but not specific cortical inhibitory neurons (*Dlx5/6*-driven targeting) are available at P2, P10, P28 and P56. For this reason, we picked P2 and P10 to represent the *immature* and early *mature* stages, respectively. Thanks to the above Reviewer's comment (1.), we have now defined time windows in Fig. 1, and we applied these labels to scRNAseq data in Fig. 3 and Supp. Fig. 3. The high expression of mGluR1 throughout maturation was confirmed by smRNA FISH and immunohistochemistry of mGluR1 at multiple time points spanning these time windows (Fig. 3f-g). We added the time window labels to these results as well and we updated the corresponding text (Lines 141-142; 152; 157).

B) Despite this, the data suggest that mGluR1 could be the candidate responsible for initiating metabotropic postsynaptic signaling in SST to regulate transient TC inputs during early postnatal development. However, the rationale behind selecting four transcription factors (TF) from the list provided in Supplementary Table 2 (*Grin3A*, *Grid1*, and two guidance molecules *Semaphorin 3A* and *7A*) is not apparent. Could the authors provide clarification on this choice?

To examine downstream effectors of mGluR1 signaling, we set a hand-picked selection of molecular candidates that were previously shown to be functionally regulated downstream of

mGluR1 (Line 193). We hypothesized that, in addition to functional protein regulation, their transcriptional levels are regulated by mGluR1. We now integrated additional references for the following (Lines 195-196).

- Grin3A has been shown by Yuan et al., 2013 to be regulated by mGluR1. In addition, Grin3A is a well-known marker of SST cells in the cortex, the basal amygdala and the hippocampus (Bossi et al., 2022), as well as during development (Murillo et al. 2021), highlighting the potential critical function of Grin3A in these cells. Moreover, the function of the receptor has recently been associated with the regulation of synaptic maturation, NMDAR-dependent plasticity and SST neuron activity (Roberts et al., 2009; Bossi et al., 2023).
- Grid1 has been shown to be activated downstream of mGluR1 in the cerebellum (Benamer et al., 2018) and is known to regulate synaptic assembly with the formation of tripartite tran-synaptic structures including Neuroxins and Cerebellins (Andrews & Dravid 2021; Fossati et al., 2019).
- *Sema3A* and *Sema7A* were investigated as potential mGluR1-downstream regulators of synaptic elimination in the cerebellum and embryonic DRG/RGC (Uesaka et al., 2014; Kreibich et al., 2004).

We found that in addition to their previously published functional interactions, mGluR1 also regulates RNA expression levels of these four gene candidates.

4. In their final figure (Figure 6), the authors examine the sequences in the Motor Sequencing platform in control and mGluR1 KD mice. They identify a set of syllables (10 syllables) that are highly enriched in the knockout (KO) animals. However, it is unclear to me how to establish a consistent link between the alteration in *Sema3A*'s effect on axon guidance and the set of syllables increased in KD mice. Could the authors provide an explanation or clarify the connection between these findings?

The Reviewer points out the complexity of Motion sequencing (MoSeq) to understand behavior-associated disruptions in our model. Reviewer #2 also commented on that point. We thank the Reviewers for these comments and agree that the behavior analysis requires clarifications.

Axon guidance and synaptic assembly have long been involved in the maturation of brain circuits and underlie the formation of proper adult behaviors. For example, previous studies suggest that BDNF and TrkB are critically involved in the mechanisms of antidepressant drugs (Castrén et al., 2005). More importantly, disruptions in semaphorin signaling (such as *Sema5A* and *Sema3F*) in mice have been associated with autism-like behaviors. However, these studies usually utilize broad gene candidate deletions, mainly using constitutive KO. The role of population-specific synaptic assembly in the maturation of functions underlying mouse behaviors is virtually unknown. Our aim is to establish whether manipulations of early developmental transients, TC inputs to SST neurons, affect the downstream maturation of the cortical functions. While PV neurons in the somatosensory cortex are known to strongly respond to active tactile behaviors, SST neurons normally control feedback inhibition (Porter et al., 2001; Cruikshank et al., 2010; Yu et al., 2019) and the integration of long-range motor and association inputs (Ietzkus et al., 2015). We therefore applied an unsupervised behavioral analysis (MoSeq) focusing on

pseudo-naturalistic behaviors of mice and took advantage of our temporal and cell-type specific CRISPR approach, in contrast to available constitutive mGluR1 KO known to exhibit ataxia.

As suggested by Reviewer #2, we improved the interpretations of the identified disruptions using MoSeq with further analyses of our data. See the following response to Reviewer #2:

- We first measured gross motor functions with velocity and total distance traveled within a session (Supp. Fig. 6), but did not identify significant defects.
- Heatmaps of mouse exploration during the recordings revealed distinct exploration patterns between Control and KD animals (Fig. 6c).
- In MoSeq experiments, the naturalistic exploration behavior in a novel, but neutral environment is parsed out in stereotyped motifs of behaviors, known as “syllables”. The set of syllables represent the grammar expressed in the exploration (body language) of each animal and each group. Each syllable is unique and identified with a number and a label for a global behavior class, such as “rear” or “groom” (Supplementary Videos). We are thankful to the Reviewer for pointing out the confusion arising from the use of global labels only in our figure. We now identified unique syllables both with their identification number and with respect to their global class.
- We added the analysis of syllable classes (“rear”, all “groom”, etc.) into integrated plots (Fig. 6d).
- Moreover, to help in the interpretation of the modifications revealed in the natural exploratory behavior unique patterns (syllables), we added exemplar videos of the syllables included in the LDA (Supp. Videos).
- Last, we modified the text in the result section (Lines 225-234) and added the above explanations in the methods section (Lines 618-623).

Using this method, we reveal that the usage of the mouse exploration grammar (syllables sequence) is modified in CRISPR-deleted animals compared to their controls. Overall, we believe Moseq data establish a correlative link between the disruption of the postnatal TC connectivity to SST neurons and the adult exploration of their environment. Further investigations are warranted to further dissect out the behaviors associated with TC network and SST neuron maturation within S1.

In summary, the manuscript poses challenges in integrating information, with a notable lack of correlation between exemplar experiments and overall values being a prominent example. The use of different ages in various experiments raises concerns about the correlation between measured activity and the degree of TC inputs. The age-related comparisons, especially in Figure 1, lack clarity. Additionally, the discrepancy between traces in Figure 2C and the quantification in Figure 2D raises questions about the conclusions drawn from this figure. The rationale behind selecting specific transcription factors (TF) in the analysis of mGluR1 signaling and the connection between alterations in Sema3A's effect on axon guidance and the syllable sequences in KD mice need further explanation. Overall, integration of information throughout the manuscript appears challenging, warranting a more coherent narrative. Further the relevance of Figure 7 is also needed (it is not commented in the text).

We thank the Reviewer for the comments, which helped us greatly improve the integration of our data in the narrative. We now added the reference to Fig. 7 (Line 266).

Reviewer #2 (Remarks to the Author):

This study brings novel insights into molecular mechanisms underlying synaptic remodeling of cortical circuits during perinatal development. Synaptic pruning is clearly a very important developmental process with mounting evidence that it is disturbed in developmental disorders, thus identifying new mechanisms is of importance.

Here the focus is on the early connection linking the thalamus to cortical somatostatin interneurons. The Fishell laboratory had previously shown that SST neurons are the first target of thalamocortical axons (TCAs) and that this connection is greatly reduced during development, with TCAs shifting their connection to parvalbumin interneurons. Using chemogenetic approach that targets selectively the SST neurons they find that that the pruning of this exuberant TCA connection is increased by the activation of excitatory DREADDs whereas inhibitory DREADDs or Ki67 have little effect. They identify the metabotropic mGluR1 receptor as a potential candidate for this: downregulation of mGluR1 in the SST neurons prevents the normal pruning of the TCA-SST connectivity. Interestingly, knockdown of mGluR1 in SST neurons downregulates the expression of a number of transcripts including a number of repellent guidance molecules such as Sema3a and sema7a which could participate in the synaptic remodeling. Finally they report that inactivation of mGluR1 in SST cells has some effects on mouse behavior that they interpret as a disruption in exploratory behavior.

The experiments are state of the art combining a number of complementary cellular, electrophysiological and molecular approaches with appropriate controls. The paper is beautifully illustrated. My comments are minor and concern some interpretations that could be reconsidered and text clarifications.

1) The specific targeting of SST neurons in all the experiments is particularly important for the clear demonstration of a role of mGluR1 in postsynaptic cells in the pruning of presynaptic afferents. In this regard it would be important that the authors place their finding in the perspective of finding in mglur1-KO mice that had indicated a somewhat similar role of mGluR1 in synapse elimination in other neural circuits such as the the elimination of climbing fibers in the cerebellum (e.g. Kano et al.1995), the lemniscal input to VB neurons (10.1371/journal.pone.0226820).

We thank the Reviewer for bringing up an interesting point. While we discussed the potential different functions of mGluR1 at different temporal stages, mGluR1 is also regionally and cell-

type specific, with distinct roles in synapse assembly, which we agree should be added to the discussion.

While mGluR1 is highly expressed by SST cINs in the immature somatosensory cortex, it is also found in other cortical cell types at adulthood and at high levels in the cerebellum and in the thalamus. mGluR1 expression is also found at high levels in the cerebellum and in the thalamus. The study of mGluR1 constitutive KO revealed temporally and regionally distinct roles for mGluR1. As pointed out by the Reviewer, multiple lines of research from Kano and colleagues, as well as the study from Narushima et al., (2019), investigated the role of mGluR1 in synapse elimination, in the cerebellum and in the thalamus. To correlate results from cell-autonomous KD and constitutive KO models, we measured TC synaptic contact density (VGluT2+/Homer1+) onto SST neurons in a constitutive mGluR1 KO brain and found that the TC synaptic contacts were increased (see attached graph), similar to cell-type specific CRISPR mGluR1 KD. This suggests that mGluR1 synaptic elimination observed in the constitutive KO includes cell-autonomous effects. Remarkably, in the previous studies using the KO, while mGluR1 was shown to promote synapse elimination, this process led to synaptic strengthening and synapse maintenance, rather than the weakening we observed with transient connectivity using CRISPR KD approach. This suggests that mGluR1 functions are highly contextualized based on both regional and temporal contexts. Furthermore, CRISPR KD approach allowed us to study behavioral consequences of cell-specific mGluR1 functions, as the constitutive KO mice display strong ataxia and important motor deficits, due to the cerebellum-associated functions of mGluR1, which prevent any behavioral study. We now expanded our discussion with the above considerations (Lines 286-302).

2) In much of the text, the authors refer to the TCA-SST connection as being transient. However from the data presented in the present study, and previous papers of the team, and other studies, the TCA-SST connection is still present in mature circuits. So this is more a remodeling process than a complete elimination of a connection, or a transient connection. Terminology should be adapted to match better this fact.

We agree with the Reviewer that the word “transient” usually refers to the elimination of an event. Nonetheless, we use “transient” to refer to the dynamics of TC inputs onto SST neurons for consistency in the field. Indeed, developmental studies of this specific network, including works from Butt, Kanold, Molnar and colleagues (review on transient cortical circuits: Molnar, Luhmann and Kanold 2020), the Tuncdemir et al., 2016 and recent work from Ghezzi et al., 2023, designate TC inputs to SST neurons as transient. We now discuss this terminology and its limitations in the discussion (Line 267-279), according to the following:

In development studies, TC inputs to SST neurons during development are referred as transient, as they are thought to be absent at adulthood, based on electrophysiological recordings (Sermet et al., 2019; Audette et al., 2018; Cruikshank et al., 2010; Tuncdemir et al., 2016). However, anatomical studies (Pouchelon et al., 2021; Wu et al., 2023; Ahrlund-Richter et al., 2019) reveal robust thalamocortical connectivity to the SST neurons at adulthood, suggesting that TC synapses are not completely eliminated after maturation and that remaining synaptic contacts are

silent, with respect to fast glutamatergic neurotransmission. Therefore, “transient” connectivity only indicates physiological dynamics of the network, as measured with electrophysiology and as compared to other cell types. The mechanisms we uncover here reveal the discrepancy between the physiological and the morphological connectivity.

We thank the Reviewer for raising this concern, as we now discuss this interesting point in our manuscript.

3) To evaluate synaptic connectivity the study relies largely on morphological analyses that do not have the necessary resolution to qualify synapses. Although Vglut2/Homer1 is a convenient proxy to estimate synaptic input close to a neuron, these are still synapse-like boutons, as stated in the material and method section, which could be stated upfront in the result section. In the mat and met section please indicate the 'thickness' of confocal plane analysed.

We thank the Reviewer for pointing out that clarifications on the morphological analyses are needed in the results section, in addition to the rest of the manuscript.

- a) First, we changed the result description from *synapse* to *synaptic contact* throughout the manuscript and added the labels “*synaptic contact density*” within the figure. The thickness of the confocal plane, of 0.0624um, was defined by the pinhole applied to the images and is now added to the method section.
- b) As suggested by Reviewer 3, we also performed further analysis using pan-AMPA labeling on the labeling of synaptic contacts to examine “silent” vs “active” synapses. These preliminary data suggest that synaptic contact we measure could also reflect, in parts, silent synapses that are not pruned. This reveals a more complex picture of the link between morphological and functional synaptic assembly. We now mention this in the result section (Line 116) and discuss anatomical versus functional synaptic maturation in the discussion (Lines 264-276).

4) The significance of changes in behaviour are difficult to understand and even more to interpret. The same behaviours (e.g. "rear up") seem alternatively as increased or decreased in the KD animals (figure 6c). It seems therefore difficult to qualify these as disruption in exploration. More standard tests of exploration should be used to substantiate this interpretation.

Similar to Reviewer #1, the Reviewer points out the complexity of Motion sequencing (MoSeq) to understand behavior-associated alterations in our model. We thank the Reviewers for these comments and agree that the behavior analysis requires clarifications on result interpretations.

Our aim is to elucidate whether manipulations of early developmental transients, TC inputs to SST neurons, affect the downstream maturation of the cortical functions. We take advantage of our temporal and cell-type specific CRISPR approach, in contrast to available constitutive KO, known to exhibit ataxia. In contrast to PV neurons in the somatosensory cortex that are known to strongly respond to active tactile behaviors, SST neurons normally control feedback inhibition (Porter et al., 2001; Cruikshank et al., 2010; Yu et al., 2019) and the integration of long-range motor and association inputs (letzkus et al., 2015). While other more standard behavioral tasks exist (novel object recognition, gap crossing tasks, different types of mazes, etc), these analyses are designed for the study of specific brain functions, which are not yet identified with respect to cell-type specific manipulations of SST neurons. To date, such CRISPR-approach enabling

regional, temporal and cell-type specificity, has not been tested with behavioral analysis. The identification of the proper behavioral task reflecting cell-type specific developmental perturbations would require to perform an extensive battery of behavioral tests, beyond the scope of this study. We therefore took a more unsupervised approach (MoSeq) focusing on pseudo-naturalistic behaviors of mice, to examine the role of the early metabotropic signaling in developing SST neurons for their adult functions.

To improve the interpretations of the disruptions that we identified using MoSeq, we performed further analysis of our data.

- We first measured gross motor functions with velocity and total distance traveled within a session (Supp. Fig. 6), but did not identify significant defects.
- Heatmaps of mouse exploration during the recordings revealed distinct exploration patterns between Control and KD animals (Fig. 6c).
- In MoSeq experiments, the naturalistic exploration behavior in a novel, but neutral environment is parsed out in stereotyped motifs of behaviors, known as “syllables”. The set of syllables represent the grammar expressed in the exploration (body language) of each animal and each group. Each syllable is unique and identified with a number and a global behavior class label, such as “rear” or “groom” (Supplementary Videos). We are thankful to the Reviewer for pointing out the confusion arising from the use of global labels. We now identified unique syllables with respect to their global class.
- We added the analysis of syllable classes (“rear”, “groom”, etc.) into integrated plots (Fig. 6d).
- Moreover, to help in the interpretation of the disruptions in the unique exploration patterns (syllable), we added exemplar videos of the syllables included in the LDA (Supp. Videos).
- Last, we modified the text in the result section (Lines 225-234) and added the above explanations in the methods section (Lines 618-623). *(Screenshot examples of the updated Fig. 6)*

Using this method, we reveal that the usage of the mouse exploration grammar (syllables sequence) is modified in CRISPR-deleted animals compared to their controls. Overall, we believe Moseq data establish a correlative link between the disruption of the postnatal TC connectivity to SST neurons and the adult exploration of their environment. Further investigations are warranted in the future to further dissect out the behaviors associated with TC network and SST neuron maturation within S1.

5) The discussion could be streamlined on the main topic of synaptic remodeling, and interaction of activity with guidance molecules in this process.

We agree with the Reviewer that the discussion of synaptic remodeling and activity-dependent guidance molecules, which was found only in the last paragraph of the discussion, could be optimized. We brought the paragraph higher in the section and expanded it with the discussion of the previous knowledge gained from constitutive mGluR1 KO (the known effects of pruning in other structures, mainly the thalamus, as suggested above), as well as the consideration of anatomical and functional aspects of transient connectivity (Lines 267-302).

Reviewer #3 (Remarks to the Author):

In the present study, the authors characterized transient thalamocortical inputs in the developing mouse sensory cortex. They confirmed that thalamocortical inputs from the VB transiently project to somatostatin (SST) inhibitory cortical interneurons (cINs) during early postnatal development. They found that enhancing the postsynaptic activity of SST cINs employing DREADD-Gq promoted the weakening of the TC inputs to SST cINs whereas reducing the postsynaptic activity by DREADD-Gi or by expressing Kir2.1 in SST cINs had almost no effects. They showed that the type 1 metabotropic glutamate receptor (mGluR1) was highly expressed in SST-cINs during development and found that CRISPR-mediated deletion of mGluR1 from SST-cINs prevented the weakening of the transient TC inputs to SST-cINs. Among candidate genes that are regulated by mGluR1 in SST-cINs based on their single-cell RNA sequencing analyses, they identified Semaphorin 3A as a molecule that mediates the effect of mGluR1 to weaken the transient TC inputs to SST-Ins. Moreover, by using the Motion Sequencing platform, they showed that mice with mGluR1 deletion in SST-cINs exhibited impairment of parameters related to exploratory behaviors. The authors argue that mGluR1 in SST-cINs regulates the weakening of TC inputs to these neurons during early postnatal development and this developmental synaptic refinement is required for the development of exploratory behavior in adult mice.

The authors used several state-of-the-art techniques to draw their conclusions and the data appear convincing. The results are very interesting and may attract readers of various neuroscience fields. I have several comments and requests as listed below.

Major Comments

1. There is a clear discrepancy between the electrophysiological data for EPSCs (Figure 2d) and morphological data for synapse density (Figure 2e). While DREADD-Gi did not affect the EPSC amplitude and charge compared to control (Figure 2d), it significantly increased the synapse density (Figure 2e). Moreover, while DREADD-Gq markedly reduced the EPSC amplitude and charge compared to control (Figure 2d), it did not affect the synapse density (Figure 2e). It seems possible that silent synapses with no postsynaptic AMPA receptors may increase by DREADD-Gi and that functional synapses may become silent while keeping synaptic structures by DREADD-Gq. To check this possibility, the authors may try immunostaining of AMPA receptors and check whether the density of VGluT2 puncta that does not colocalize with AMPA receptor signal changed after DREADD-Gq and DREADD-Gi.

We are extra-thankful to the Reviewer's comment, as they gave us an opportunity to inspect the data more thoroughly and to identify an unfortunate mistake. A batch of P30 data, instead of the P10 control dataset, was erroneously used in our analysis. This was at the origin of the unexpected discrepancy between these data and the rest of the manuscript. We now corrected this mistake (see screenshot of updated Fig. 2e).

Nonetheless, we thought the Reviewer raised an interesting point, suggesting the presence of silent synapses. We thought of interest to further investigate morphological synaptic assembly by examining “silent synapses” with AMPA receptor staining in our datasets, as classical Hebbian development involves plasticity from silent synapses (Feldman et al., 1999; Isaac et al., 1997; Faust et al., 2021). We found that around half of VGluT2(+) synaptic boutons are colocalized with postsynaptic pan-AMPA receptors with no significant difference with SST neurons expressing DREADD-Gi. (CTRL : 44.81 +/- 4.96% ; Gi : 78.70 +/-

5.73 % Mean±SEM). While not regulated by hyperpolarization, this suggests the presence of silent synapses during the remodeling of transient connectivity. We did not add these preliminary data to the manuscript as this issue was primarily driven by our regrettable mistake and they now only raised open-ended questions rather than adding to the current interpretations.

Additionally, we modified the figures and the text by replacing “synapses density” by “synaptic contact density”, as suggested by Reviewer #2. Overall, the synaptic assembly associated with transient connectivity is largely overlooked and a comprehensive understanding of the process would warrant a separate study, ideally using high-throughput 3D electron microscopy (for example Gour et al., 2020; Wildenberg et al., 2023). This is now discussed in our manuscript (Lines 267-302)

2. Throughout the manuscript, the authors describe that the transient connectivity from the VB to SST-cINs inversely correlated with “postsynaptic neuron activity.” This argument is mainly based on the results of the DREADD-Gq experiment (Figure 2, Supplementary Figure 2). However, the activation of DREADD-Gq induces not only depolarization leading to action potential generation but also driving Gq signaling cascade that may not accompany membrane depolarization. Since mGluR1 mediates the developmental weakening of TC inputs to SST-cINs, the activation of DREADD-Gq is considered to bypass mGluR1 and directly activate the Gq-signaling cascade downstream of mGluR1. It is therefore unclear whether the depolarization of SST-cINs promotes the weakening of TC inputs. The failure to induce the reduction of the EPSC amplitude and charge by the Kir2.1 expression (Supplementary Figure 2f, g) suggests that the membrane potential of

postsynaptic SST-cINs does not play a major role. Since continuous optogenetic manipulation of SST-cINs in developing mouse pups in vivo is difficult, the authors may check whether the DREADD-Gq-induced promotion of synaptic weakening is prevented by the expression of Kir2.1 in SST-cINs.

We agree with the Reviewer that the mechanisms involved in the Gq signaling are not entirely clear. In the canonical Hebbian system, as classically described during network development (Isaac et al., 1999), TC synapses become strengthened through correlated pre and postsynaptic activity and long-term potentiation (LTP). However, our results reveal that the early TC synapses to SST neurons are instead weakened. It is of interest to understand whether DREADD-Gq triggers signaling cascade alone or depolarization of SST neurons also regulates the TC input weakening.

We thank the Reviewer for allowing us to perform this investigation, we now refer to the excitability of SST neurons in the result section and corrected “inversely correlated with postsynaptic activity, for “non-canonical” and “metabotropic signaling-driven” mechanisms . Indeed, several lines of evidence support the Reviewer’s hypothesis that depolarization or hyperpolarization is not directly involved in the acute weakening of TC inputs onto SST neurons. (Abstract Line 28; Results: Lines 128-131)

a) The excitability of SST INs manipulated with DREADD-Gq is not affected (Table S1)

b) The excitability of SST INs in mGluR1 KD is not affected (Table S4).

c) Contrarily, the constitutive hyperpolarization of SST neurons with Kir2.1 did not affect the TC connectivity (Supplementary Fig. 2g-i).

e) The excitability of SST INs is partly modified by DREADD-Gi (Table S1) but did not affect TC connectivity either (Fig. 2c-d).

d) In a separate set of experiments (data not shown), increasing depolarization of developing SST neuron excitability (with the expression of the sodium channel NaChBach, known to enhance neuron activity) did not recapitulate the effect we observed on transient connectivity using DREADD-Gq. Experimental design: AAV-Flex-NaChBac injected at P0, optogenetic stimulation of TC inputs at P21 and measure of SST neuron responses (EPSC amplitudes, Mean±SEM): Ctrl: 296.8±62.80 n=15; Kir2.1: 84.76±29.15 n=8; NaChBac: 563.3±204 n=13, Non-significant Kruskal-Wallis test p=0.1149, followed by Dunn’s multiple comparison: Ctrl vs Kir adj. p=0.93; Ctrl vs NaChBac adj. p>0.99; Kir vs NaChBac adj. p=0.27).

3. Supplementary Figure 2f, g: The authors check whether the synapse density increases in Kir2.1-expressing SST-cINs similarly to SST-cINs in which DREADD-Gi was activated (Figure 2e).

4. Figure 5f-h: The authors should examine whether the synapse density is increased in Sema 3A-KD SST-cINs as they examined in mGluR1-KD SST-cINs.

As discussed above (major comment 1), synaptic contact refinement potentially does not involve the same mechanisms as the functional maturation of the synapses. We had measured synaptic contact density onto SST neurons expressing Kir2,1 in a separate set of experiments at P30. Similar to SST neurons manipulated with DREADD-Gi, no significant effect was detected. We now added these data in Supp. Fig. 2i).

Since electrophysiological recordings are more robust throughout all manipulations and typically define transient connectivity, we believe that they are more appropriate and informative readouts of each manipulation beyond the developmental data.

Minor points

1. Page 6, lines 118-120 “After CNO discontinuation (P8), these effects proved temporary, as TCVB inputs were not reduced onto Gq(+) SST cINs anymore by P30 (Supplementary Fig. 2e).” The EPSC amplitude of Gq at P30 (57.38 ± 11.75 , $n = 17$, Supplementary Fig. 2e) seems significantly larger than Gq at P10 (12.68 ± 2.021 , $n = 8$, $N = 3$, Fig. 2d). Does this mean that TC inputs to SST-cINs increase after the termination of DREADD-Gq activation while the TC inputs decreased during the same developmental period in control mice? Please comment on this point.

The amplitude of SST neuron responses evoked by TC stimulation at P30, 22 days after discontinuation of CNO show a mild but significant increase between Ctrl and DREADD-Gq SST neurons. In order to further investigate this effect, we added the quantification of the EPSC charge (pC) to this analysis, which is not significantly affected (Fig. 2f). It is possible that the artificial enhancement of metabotropic-dependent pathways during development leads to a mild rebound effects or that the long-term over-expression of engineered receptors using AAVs, even without CNO activation, triggers membrane or excitability effects, as normal duration of AAV infection is usually shorter. We now comment this in the result section (Line 122).

2. Figure 2c: The sample EPSC trace of Gi in Fig. 2c is not representative since DREADD-Gi activation did not change the EPSC amplitude and charge (Fig. 2d).

We thank the Reviewer, who, like Reviewer #1, pointed out the confusion arising from the traces representing the results. We now updated Fig. 2c with more representative traces (average of unitary example cells) picked from the same dataset.

3. In Figure 6, the data for mGluR1-KD mice are highlighted in bright yellow, but it is rather difficult to see. Furthermore, the letters in Fig. 6c are too small to read. Please use a different color and make the letters larger.

Upon the modification of Fig. 6 based on the Reviewers' comments, we also modified the layout of the figure.

4. Supplementary Fig.2g: In the legend for Supplementary Fig.2g, the values for “Ctrl” are not shown.

This is now corrected, with the addition of the Ctrl values.

REVIEWER COMMENTS

Reviewer #1 (Remarks to the Author):

These are my comments to the revised version:

1. Concerning the first point, the authors have categorized the results as immature based on previously published findings, up to P7, when no physiological weakening has occurred. There are two immature windows: one in the range of P9-P11, marked as the early mature stage, and the other in the P28-P32 range. These points are now clarified in the legend.
2. The second point was regarding suboptimal traces in Fig. 2d (old version). The examples chosen now are suitable for the information gathered from the entire Figure 2.
3. The idea of this point was to explore how the authors chose Grin3A, Grid1, and two guidance molecules, Semaphorin 3A and 7A. Now, the authors explain that they selected a hand-picked set of molecular candidates, previously shown to be functionally regulated downstream of mGluR1. The reasons for such choices are now explained in the text.
4. In the previous version of the manuscript, the last figure was presented without a clear introduction to the possible relationship between the set of syllables highly enriched in the knockout animals. In the last version, there is an extended and clearer presentation of the possible interpretation of the syllables.

Overall, I believe that this revised version of the manuscript is more robust than the previous one and, in my view, is ready for publication.

Reviewer #2 (Remarks to the Author):

The authors have adequately responded to the points raised by the reviewers and improved the presentation of their data in particular regarding behavioral analysis. They have clarified some methodological details and provide a more balanced interpretation of their results.

Reviewer #3 (Remarks to the Author):

The authors have addressed essentially all my comments satisfactorily and improved their manuscript accordingly. I have no further comments.